

# Revision of the Late Jurassic deep-water teleosauroid crocodylomorph *Teleosaurus megarhinus* Hulke, 1871 and evidence of pelagic adaptations in Teleosauroidea

Davide Foffa[1,2,*], Michela M. Johnson[2,*], Mark T. Young[2,*], Lorna Steel[3] and Stephen L. Brusatte[1,2]

[1] Current affiliation: Department of Natural Sciences, National Museums Scotland, Edinburgh, United Kingdom
[2] School of GeoSciences, University of Edinburgh, Edinburgh, United Kingdom
[3] Department of Earth Sciences, Natural History Museum, London, United Kingdom
[*] These authors contributed equally to this work.

Corresponding author
Davide Foffa, D.Foffa@nms.ac.uk, davidefoffa@gmail.com

## ABSTRACT

Teleosauroids were a successful group of semi-aquatic crocodylomorphs that were an integral part of coastal marine/lagoonal faunas during the Jurassic. Their fossil record suggests that the group declined in diversity and abundance in deep water deposits during the Late Jurassic. One of the few known teleosauroid species from the deeper water horizons of the well-known Kimmeridge Clay Formation is '*Teleosaurus*' *megarhinus* Hulke, 1871, a poorly studied, gracile longirostrine form. The holotype is an incomplete snout from the *Aulacostephanus autissiodorensis* Sub-Boreal ammonite Zone of Kimmeridge, England. The only other referred specimen is an almost complete skull from the slightly older *A. eudoxus* Sub-Boreal ammonite Zone of Quercy, France. Recently, the validity of this species has been called into question. Here we re-describe the holotype as well as the referred French specimen and another incomplete teleosauroid, DORCM G.05067i-v (an anterior rostrum with three osteoderms and an isolated tooth crown), from the same horizon and locality as the holotype. We demonstrate that all specimens are referable to '*Teleosaurus*' *megarhinus* and that the species is indeed a valid taxon, which we assign to a new monotypic genus, *Bathysuchus*. In our phylogenetic analysis, the latest iteration of the ongoing Crocodylomorph SuperMatrix Project, *Bathysuchus megarhinus* is found as sister taxon to *Aeolodon priscus* within a subclade containing *Mycterosuchus nasutus* and *Teleosaurus cadomensis*. Notably *Bathysuchus* has an extreme reduction in dermatocranial ornamentation and osteoderm size, thickness and ornamentation. These features are mirrored in *Aeolodon priscus*, a species with a well-preserved post-cranial skeleton and a similar shallow and inconspicuous dermal ornamentation. Based on these morphological features, and sedimentological evidence, we hypothesise that the *Bathysuchus* + *Aeolodon* clade is the first known teleosauroid lineage that evolved a more pelagic lifestyle.

## INTRODUCTION

Teleosauroids (one of the subgroups of Thalattosuchia) were a successful group of semi-aquatic Jurassic crocodylomorphs. They were abundant in marine/lagoonal faunas for most of the Jurassic of Europe, Asia and Africa (*Andrews, 1909*; *Andrews, 1913*; *Buffetaut, Termier & Termier, 1981*; *Lepage et al., 2008*; *Young et al., 2016*; *Jouve et al., 2016*; *Johnson et al., 2018*), and continued into the Early Cretaceous of Africa (*Fanti et al., 2016*). Teleosauroids underwent a severe decline across the Middle-Late Jurassic boundary at Sub-Boreal and Boreal latitudes, but remained numerically and taxonomically abundant in the Late Jurassic of the Tethys and continental Europe (*Young et al., 2014a*; *Young et al., 2014b*; *Foffa, Young & Brusatte, 2015*; *Johnson et al., 2015*; *Johnson et al., 2017*).

The teleosauroid fossil record is particularly sparse in the Late Jurassic Kimmeridge Clay Formation (KCF; Kimmeridgian-Tithonian, ∼157–148 Ma) of the UK (*Young & Steel, 2014*), where rare fossilised remains are almost exclusively limited to isolated tooth crowns (NHMUK PV R 1774) and osteoderms (MJML K2158 and BRSMG Ce9826; CAMSM J.29481; OUMNH J.77970-1) (*Seeley, 1869*; *Young & Steel, 2014*; *Foffa, Young & Brusatte, 2018*). The only exceptions are the cranial remains of an enigmatic longirostrine teleosauroid, '*Teleosaurus*' *megarhinus*, from the deep-water deposits of the Kimmeridgian of Dorset (England, UK) and a shallower (up to 100 m) contemporaneous formation of Franculès (Quercy, France) (*Hulke, 1871*; *Delair, 1958*; *Vignaud et al., 1993*). The lack of information on this taxon is frustrating, as it not only comes from a well-sampled interval, but it is also one of the few teleosauroids that may have adapted to deeper-water environments, as opposed to the more nearshore ecosystems that most members of the clade inhabited. Thus, '*Teleosaurus*' *megarhinus* may provide pivotal insights into the palaeobiology and evolution of teleosauroids.

Here we review the two previously identified specimens of '*Teleosaurus*' *megarhinus* (*Hulke, 1871*), and describe a new specimen that can also be referred to this taxon (DORCM G.05067i-v). Our description of this material reveals a unique combination of characters that validates the species. With the aid of an updated and expanded teleosauroid phylogenetic dataset, we test the relationships of this species, but find that it does not group with the type species of *Teleosaurus*, *T. cadomensis*, but rather within a clade that includes another supposed deep-water teleosauroid. This necessitates the establishment of a new genus name for '*Teleosaurus*' *megarhinus*, and indicates that there was a subclade of teleosauroids adapted to a deeper-water, open-ocean environment during the Jurassic. Along with their purely pelagic cousins, the metriorhynchids, these teleosauroids were an independent thalattosuchian invasion into the marine realm during the Jurassic, revealing previously unrecognized parallelism within this major group of early crocodylomorphs.

### Historical background

The holotype of *Teleosaurus megarhinus* (NHMUK PV OR 43086) was discovered in the winter of 1870 in the Kimmeridge Clay Formation strata at Kimmeridge Bay (Dorset), in England. J.C. Mansel-Pleydell (a Dorset antiquary, famous for his contribution to geology, botany and zoology) sent the specimen to J.W. Hulke to be described (*Hulke, 1871*; *Delair, 1958*). After making comparisons with other specimens, Hulke assigned the specimen

to *Teleosaurus* (without supporting the decision) and proposed the specific designation *megarhinus* based on the 'dilation of the terminal nostril' that he considered as 'greater than in any other *Teleosaurus* known to me' (sic *Hulke, 1871*, pp. 442). *Lydekker (1888)* later referred the species to the genus *Steneosaurus*.

A skull from near Quercy (France) was referred to '*Steneosaurus*' cf. *megarhinus* based on its total skull (premaxillary and maxillary) tooth count, and stratigraphic occurrence (*Vignaud et al., 1993*). *Vignaud et al. (1993)* also compared '*S.*' *megarhinus* with other longirostrine teleosauroids from the Late Jurassic of Europe (*Steneosaurus deslongchampsianus Lennier, 1887* and *Aeolodon priscus Von Sömmerring, 1814*), and considered '*S.*' *megarhinus* to be a valid taxon, but they were only able to differentiate these three species based on tooth counts. *Vignaud (1997)* also noticed differences in the dentition (tooth count and crown shape proportions) between '*Steneosaurus*' *leedsi Andrews, 1909*, and '*S.*' *megarhinus*.

*Pierce, Angielczyk & Rayfield (2009)* hypothesised that '*Steneosaurus*' *megarhinus* was a synonym of '*Steneosaurus*' *leedsi*. Under the rules of the ICZN Code this would have resulted in '*S.*' *megarhinus* being the senior subjective synonym, not '*S.*' *leedsi* (contra *Pierce, Angielczyk & Rayfield, 2009*) who considered '*S.*' *leedsi* as the senior synonym). It would also indicate a species that lasted for approximately 12 million years, and was morphologically distinct at both chronostratigraphic termini. However, the teleosauroid species diagnoses of *Pierce, Angielczyk & Rayfield (2009)* have been criticised as being diagnostic only to the generic level or using characters that describe all teleosauroids (*Martin & Vincent, 2013*, p. 194). Furthermore, *Pierce, Angielczyk & Rayfield (2009)* reported that *S. megarhinus* is known *"from the Oxfordian-Kimmeridgian of England and Germany"* (sic). However, *contra Pierce, Angielczyk & Rayfield*, (2009, p. 1067), we cannot find any mention of any German or Oxfordian specimens referred to '*S.*' *megarhinus*, by *Vignaud (1995)* or anyone else.

The validity problem of '*S.*' *megarhinus* is due to the use of overall upper jaw tooth total count as the sole means to differentiate the taxon from other teleosauroids (e.g., *Vignaud et al., 1993*; *Pierce, Angielczyk & Rayfield, 2009*). This has caused considerable confusion on the taxonomy of long snouted teleosauroids. Thus, it is not surprising that the validity of the species has been questioned. Instead, other significant features (e.g., the characteristic shape and extreme lateral expansion of the premaxilla, the arrangement of the premaxillary alveoli, and number of premaxillary alveoli) are rarely (if ever) considered in systematic studies of teleosauroids, until this study.

Only one phylogenetic analysis (*Mueller-Töwe, 2006*), so far, has included '*S.*' *megarhinus* and recovered it as the sister taxon to *Teleosaurus cadomensis Lamouroux, 1820* (the type species of *Teleosaurus*), based on prefrontal characters (*Mueller-Töwe, 2006*). This would be consistent with the initial placement of '*S.*' *megarhinus* as a species of *Teleosaurus*, as originally proposed by Hulke. Subsequently, the validity and systematics of '*Teleosaurus*' *megarhinus* have not been further investigated.

## GEOLOGICAL SETTING AND PALEOENVIRONMENT

Both NHMUK PV OR 43086 and DORCM G.05067i-v were found at the same general locality: Kimmeridge Bay (Dorset, UK), the type locality of the Kimmeridge Clay Formation (Fig. 1) (KCF; Kimmeridgian-Tithonian, ~157–148 Ma). In England, the KCF outcrops onshore from Dorset to Yorkshire, and continues offshore as one of the main source rocks for the North Sea oil industry. Other important KCF localities in the UK are found in Scotland on the western shores of the Isle of Skye (Inner Hebrides) and southern Sutherland (*McArthur, Hartley & Jolley, 2013*). The KCF comprises a succession of silicoclastic marine deposits dominated by calcareous organic-rich mudstones, claystones, and siltstones, frequently intercalated with oil-rich shales, and concretionary horizons (*Cox & Gallois, 1981*; *Gallois, 2004*). The KCF is traditionally subdivided into the Lower KCF (*Pictonia baylei* to *Aulacostephanus autissiodorensis* ammonite zones—Kimmeridgian) and Upper KCF (*Pectinates elegans* to *Virgatopavlovia fittoni* ammonite zones—early Tithonian). The KCF is part of the Ancholme Group that, spanning the Middle-Late Jurassic, offers a relatively continuous lithostratigraphic and fossil record of an epicontinental sea (Jurassic Sub-Boreal Seaway) that covered a large part of the modern British Isles at that time. The KCF strata record a long-term transgressive cycle that started in the middle Oxfordian following a regression phase in the Callovian-early Oxfordian (*Coe, 1992*; *Coe, 1995*; *Cox, 2001*; *Gallois, 2004*; *Weedon, Coe & Gallois, 2004*). The Kimmeridgian strata of the KCF record a deepening phase of the Jurassic Sub-Boreal Seaway, during a period of high global sea levels (*Cox, 2001*). The KCF strata at Kimmeridge Bay span the middle part of the Kimmeridgian stage (*Aulacostephanus eudoxus* to *Pectinatites wheatleyensis* ammonite Subzones). Thus, this section likely represents the deepest environment (outer-shelf water depth of 150–200 m) (*Gallois, 2004*) where teleosauroid fossils have been found (see Discussion).

The Upper Kimmeridgian strata of the Franculès (Quercy) area where the LPP specimen was found consist of argillaceous limestones intercalated with marls (Fig. 1). The rich ammonite fauna of the section indicates that the teleosauroid fossil come from the *A. eudoxus* ammonite Zone (specifically between the *Quercynum* Horizon [*Caletenum* Subzone] and the *Contejeani* Horizon [*Contejeani* Subzone]) (*Hantzpergue & Lafaurie, 1983*; *Hantzpergue, 1989*). The associated invertebrate fauna (an ostreid bivalve and ammonite assemblage) indicate a palaeoenvironment no deeper than 100 m (*Hantzpergue & Lafaurie, 1983*; *Hantzpergue, 1989*).

## SYSTEMATIC PALEONTOLOGY

CROCODYLOMORPHA *Hay, 1930* (*sensu Nesbitt, 2011*)

THALATTOSUCHIA *Fraas, 1901* (*sensu Young & Andrade, 2009*)

TELEOSAUROIDEA *Geoffroy Saint-Hilaire, 1831* (*sensu Young & Andrade, 2009*)

*BATHYSUCHUS*, gen. nov. (Figs. 2–8)

ZooBank Life Science Identifier (LSID) for genus urn:lsid:zoobank.org:act:5D902DD6-AE09-466D-8C40-C729F8481636

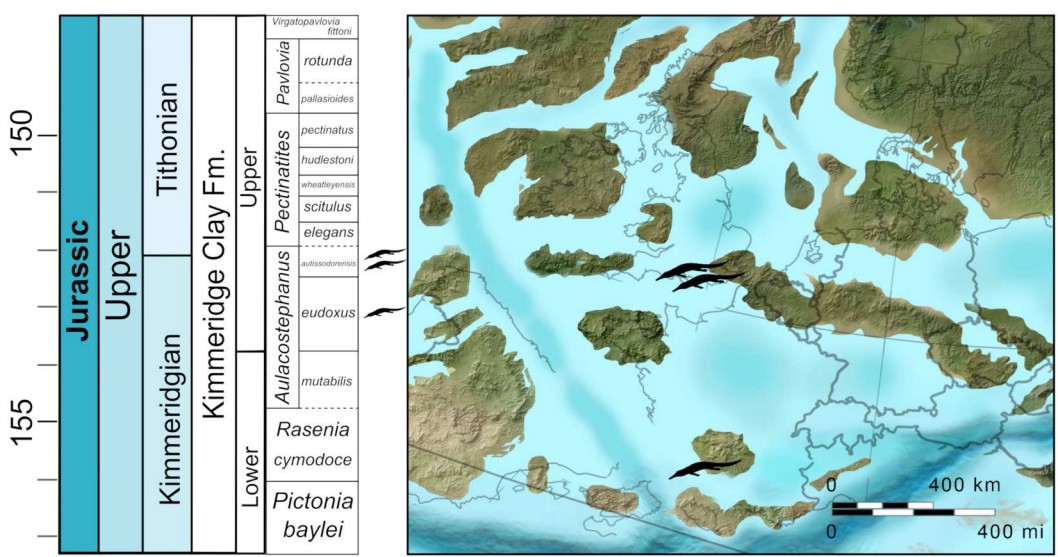

**Figure 1** Stratigraphic and palaeogeographic distribution of Late Jurassic (Kimmeridgian—early Tithonian) *Bathysuchus megarhinus* Map modified from Ron Blakey ©(http://cpgeosystems.com/).

The electronic version of this article in Portable Document Format (PDF) will represent a published work according to the International Commission on Zoological Nomenclature (ICZN), and hence the new names contained in the electronic version are effectively published under that Code from the electronic edition alone. This published work and the nomenclatural acts it contains have been registered in ZooBank, the online registration system for the ICZN. The ZooBank LSIDs (Life Science Identifiers) can be resolved and the associated information viewed through any standard web browser by appending the LSID to the prefix http://zoobank.org/. The LSID for this publication is: [urn:lsid:zoobank.org:pub:BA30BB3C-9D18-48ED-A79B-AA660450E54B]. The online version of this work is archived and available from the following digital repositories: PeerJ, PubMed Central and CLOCKSS.

**Type species**—*Bathysuchus megarhinus* gen. et comb. nov. (type by monotypy).

**Etymology**—Meaning deep water crocodile. 'βαθύς (*bathus*)' is Ancient Greek for 'deep', and—'σοῦχος (*soûkhos*)' is Ancient Greek for crocodile

**Diagnosis**—Same as for the only known species (type by monotypy).

*BATHYSUCHUS MEGARHINUS,* gen. et comb. nov.

v*1871 *Teleosaurus megarhinus* nov. sp.; Hulke, p. 442-443, pl. 18, fig.1-3

v 1872 *Steneosaurus morinicus* (sic) nov. sp.; Sauvage, p. 180

v 1874 *Steneosaurus morinicus* Sauvage; Sauvage, p. 38-40

v 1888 *Steneosaurus megarhinus* (*Hulke, 1871*)—Lydekker, p. 117

v 1936 *Steneosaurus megarhinus* (*Hulke, 1871*)—*Kuhn, 1936*, p. 39

v 1936 *Steneosaurus morinicus* (*Sauvage, 1874*)—*Kuhn, 1936*, p. 33

v 1958 *Teleosaurus megarhinus* (*Hulke, 1871*)—Delair, p. 57

v 1973 *Steneosaurus megarhinus* (*Hulke, 1871*)—*Steel, 1973*, p. 33

v 1973 *Steneosaurus morinicus* (*Sauvage, 1874*)—*Steel, 1973*, p. 32

v 1986 *Steneosaurus morinicus* (*Sauvage, 1874*)—*Buffetaut, Rose & Vadet, 1986* p. 80-81

v 1993 *Steneosaurus* cf. *megarhinus* (*Hulke, 1871*)—Vignaud et al., p.1509-1514, fig.2

v 2006 *Steneosaurus megarhinus* (*Hulke, 1871*)—Mueller-Töwe

v 2009 *Steneosaurus leedsi* (*Andrews, 1909*)—Pierce et al.

v 2012 *Steneosaurus megarhinus* (*Hulke, 1871*)—*Bronzati, Montefeltro & Langer, 2012*

**Holotype**—The specimen NHMUK PV OR 43086 is an incomplete, diagenetically damaged and partially reconstructed snout (including fragments of the anterior maxillae and posterior processes of the premaxillae) (Fig. 2). Although this specimen is partially distorted, after careful examination of the specimen (as well as more deformed specimens from closely related species) we consider the deformation on the holotype to be minimal. Crucially the morphology of the premaxillae is considered genuine and is consistent with the other referred—and less distorted—specimens.

**Referred specimens**—DORCM G.05067i-v is an incomplete but well-preserved and minimally distorted snout (i) (Fig. 3), including most of the premaxillae and a limited portion of the anterior parts of the maxillae, which was found at the same locality and in the same horizon as the type species. A complete tooth (v) (Fig. 7) and three well preserved osteoderms (ii–iv) (Fig. 8) were also found associated with the snout.

Another specimen from Francoulès, Quercy area, France (*A. eudoxus* ammonite Zone) was referred to as '*Steneosaurus*' cf. *megarhinus* and is now housed at the Université de Poitiers (LPP) (Figs. 4–6). *Vignaud et al. (1993)* described this specimen, which consists of: two skull pieces, one being the rostrum and the second the posterior skull (Figs. 4 and 5); and the anterior portion of the mandibular symphysis, posteriorly broken between alveoli D28 and D29 (Fig. 6). The rostrum is well preserved; however, the skull is broken and displaced and several parts are replaced by plaster. This makes the sutures of the dorsal surface nearly indistinguishable. The orbits appear circular/subcircular in dorsal view (Fig. 5); the anteromedial margins of both orbits have been dorsally displaced across a fracture that has dorsally displaced the entire postorbital skull. Thus, all available information pertaining to the posterior cranium and lower jaw of *Bathysuchus* comes from this specimen. *Vignaud et*

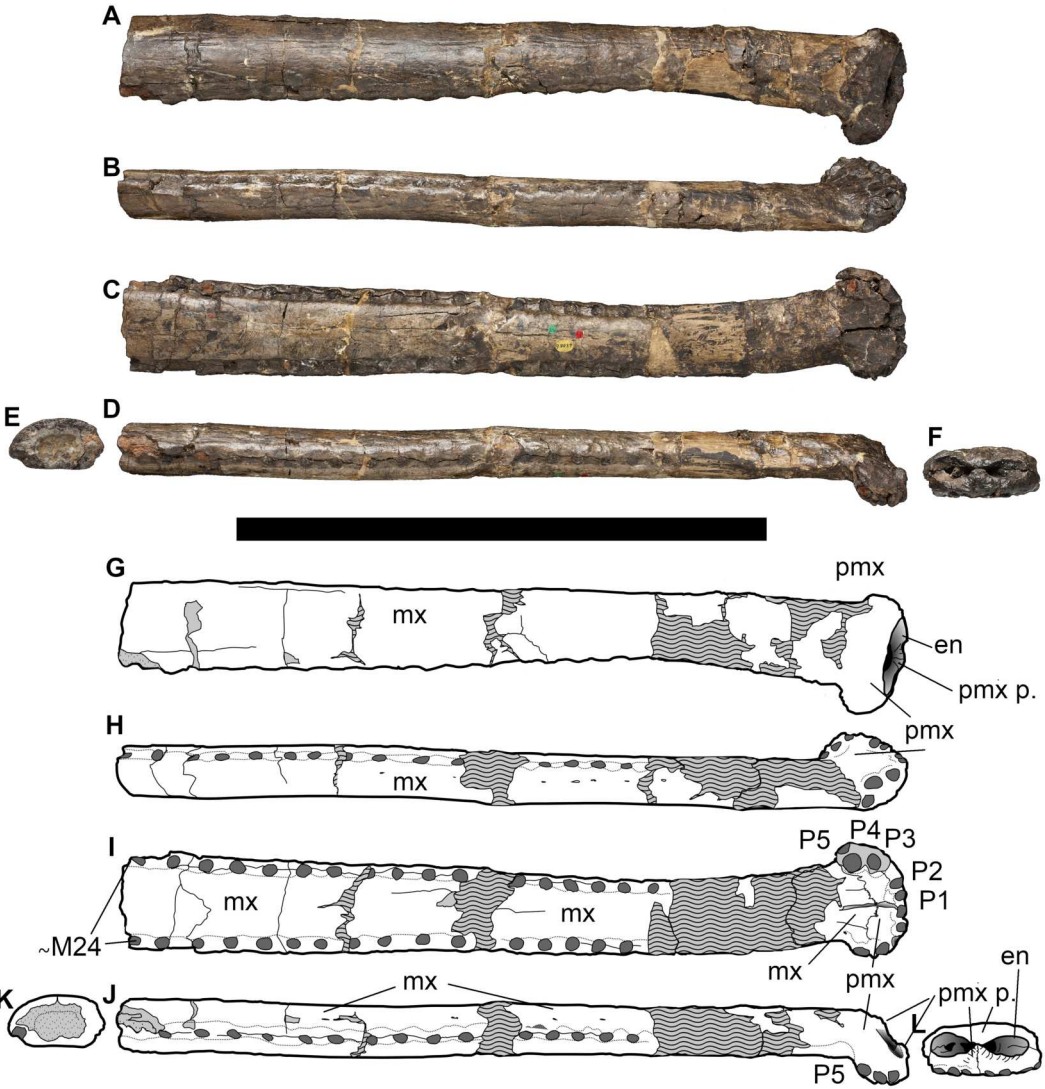

**Figure 2** **NHMUK PV OR 43086, holotype of *Bathysuchus megarhinus* gen. et. sp. nov. from the Kimmeridgian of Kimmeridge Bay, Dorset, UK, and interpretative drawings.** A, G, middle and anterior rostrum in dorsal view. B, H, rostrum in left lateral view. C, I, rostrum in ventral view. D, J, rostrum in right lateral ventral view. E, K, maxillae in posterior view. F, L, premaxilla in anterior view. Wave pattern represents parts of the specimen that have been reconstructed during preparation/conservation. Scale bar equals 30 cm. ©The Trustees of the Natural History Museum, London.

*al. (1993)* considered this specimen to be a juvenile based on the smooth ornamentation of the dermatocranium, and the unclosed postfrontal-frontal and nasal-frontal sutures. While we agree that the specimen is probably immature (based on size comparisons with other referred taxa), we also argue that the above-mentioned features cannot be convincingly used to support this conclusion (see Discussion).

**Other material**—*Vignaud (1995)* also referred an anterior fragment of the mandible (BHN 2R 25; the holotype of '*Steneosaurus*' *morinicus* (*Sauvage, 1872*) as '*T*'. *megarhinus*. We
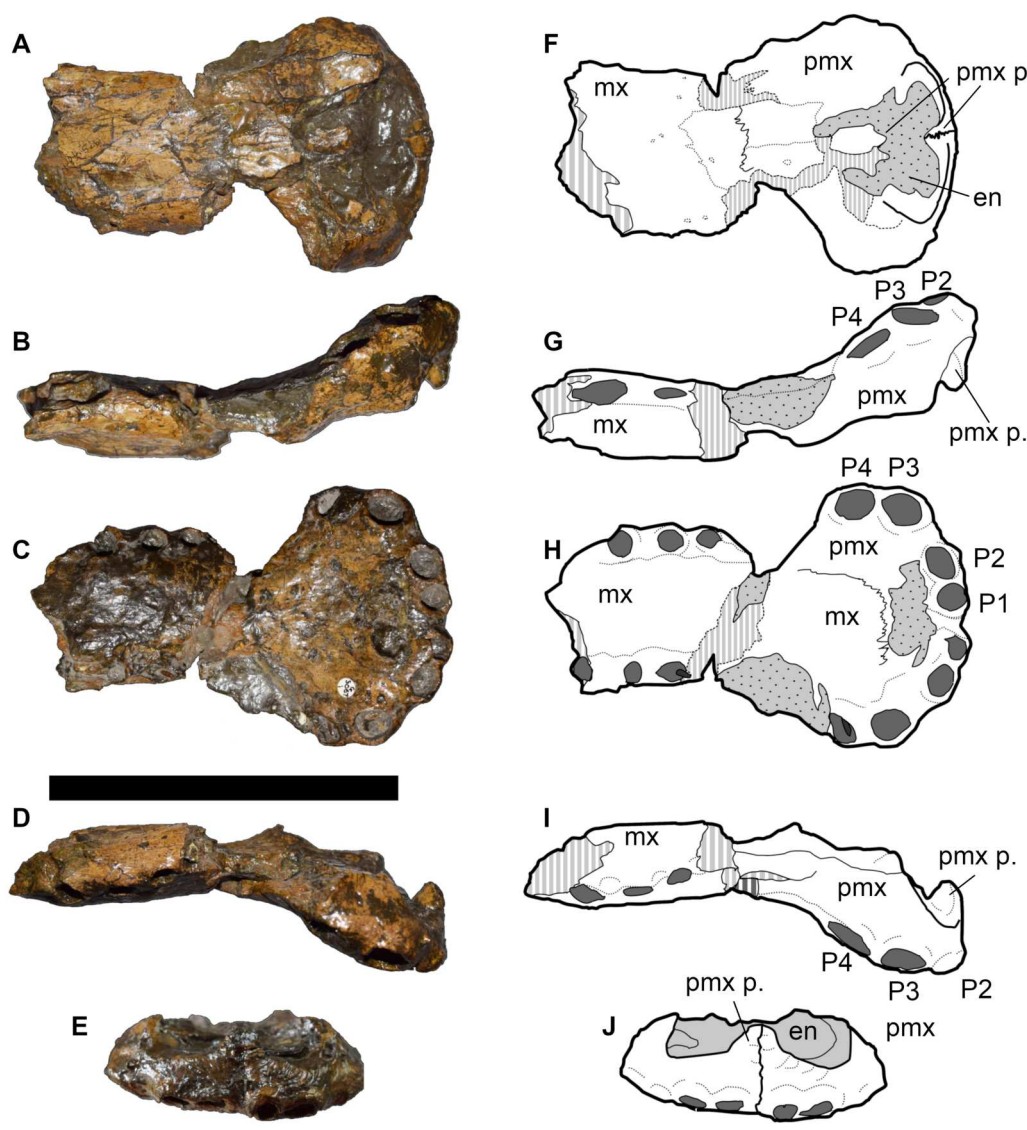

**Figure 3** DORCM G.05067i, anterior rostrum of referred specimen of *Bathysuchus megarhinus* gen. et. sp. nov. from the Kimmeridgian of Kimmeridge Bay, Dorset, UK, and interpretative drawings. A, F, anterior maxillae and premaxillae in left dorsal view. B, G, anterior maxillae and premaxillae in left lateral view. C, H, anterior maxillae and premaxillae in ventral view. D, I, anterior maxillae and premaxillae in in right lateral view. E, J, premaxilla in anterior view. Scale bar equals 10 cm.

have not seen this specimen (composed of the anterior mandibular symphysis and two associated vertebrae) and, based on information in the literature, we cannot confidently comment on its affinities.

**Type horizon**—*A. autissiodorensis* ammonite Zone, Kimmeridge Clay Formation.

**Diagnosis**—Longirostine teleosauroid (rostrum 71% of total basicranial length) crocodylomorph with the following unique combination of characters among thalattosuchians (autapomorphic characters are indicated by an asterisk*): the premaxillae

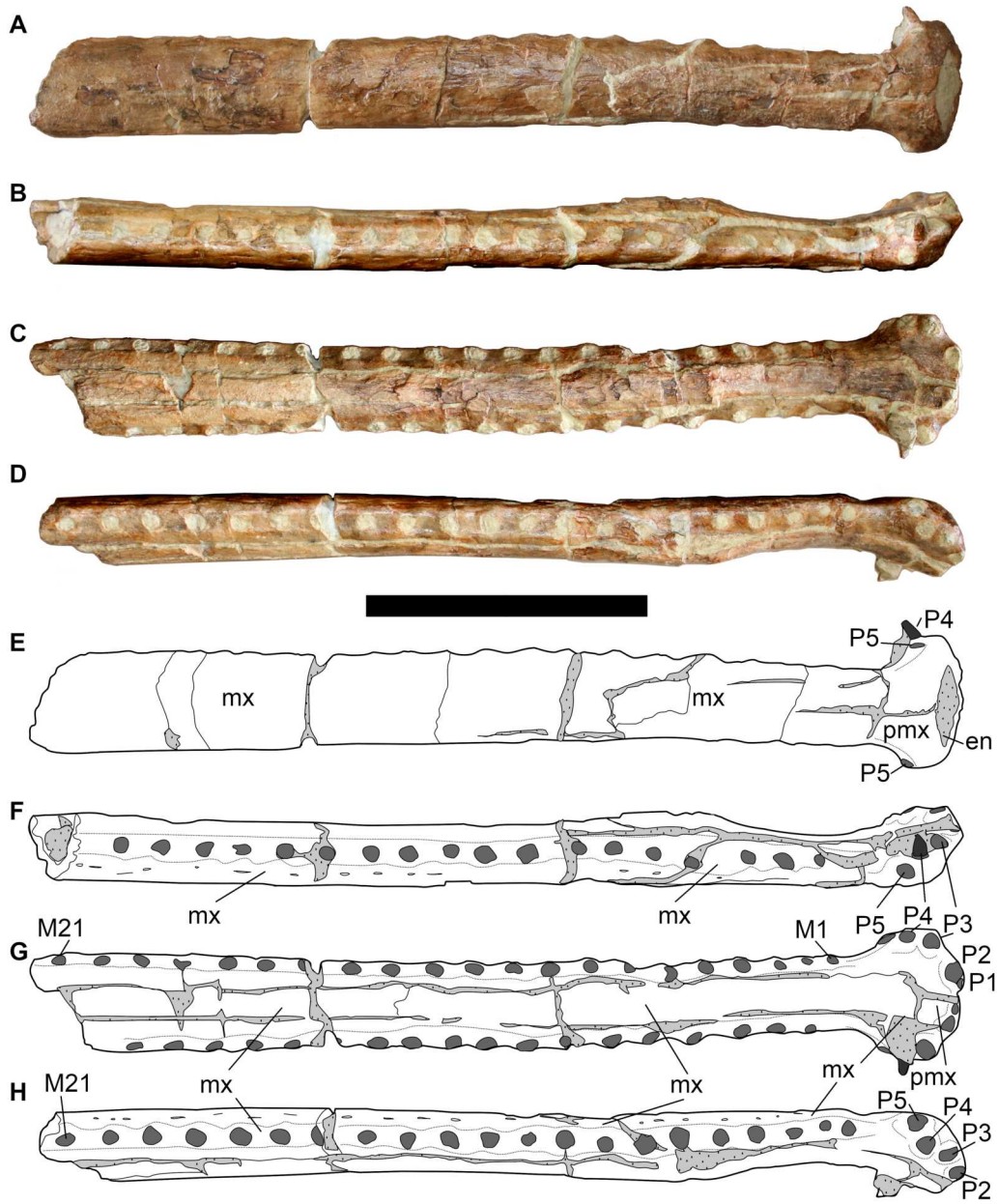

**Figure 4** **LPP specimen, anterior rostrum of referred specimen of *Bathysuchus megarhinus* gen. et. sp. nov. from the Upper Kimmeridgian of Franculés, Quercy, France, and interpretative drawings.** A, E, anterior maxillae and premaxillae in dorsal view. B, F, anterior maxillae and premaxillae in left lateral view. C, G, anterior maxillae and premaxillae in ventral view. D, H, anterior maxillae and premaxillae in right lateral view. Scale bar equals 10 cm.

have five alveoli (shared with *Platysuchus multiscrobiculatus* *Berckhemer, 1929*, *Teleosaurus cadomensis* *Lamouroux, 1820*; '*Steneosaurus' jugleri* nomen dubium (*Von Meyer, 1845*); '*Steneosaurus' deslongchampsianus* (*Lennier, 1887*; *Savalle, 1876*)); the P1 and P2 alveoli are lateral to each other at the anterior margin of the premaxilla (shared with *Mycterosuchus*

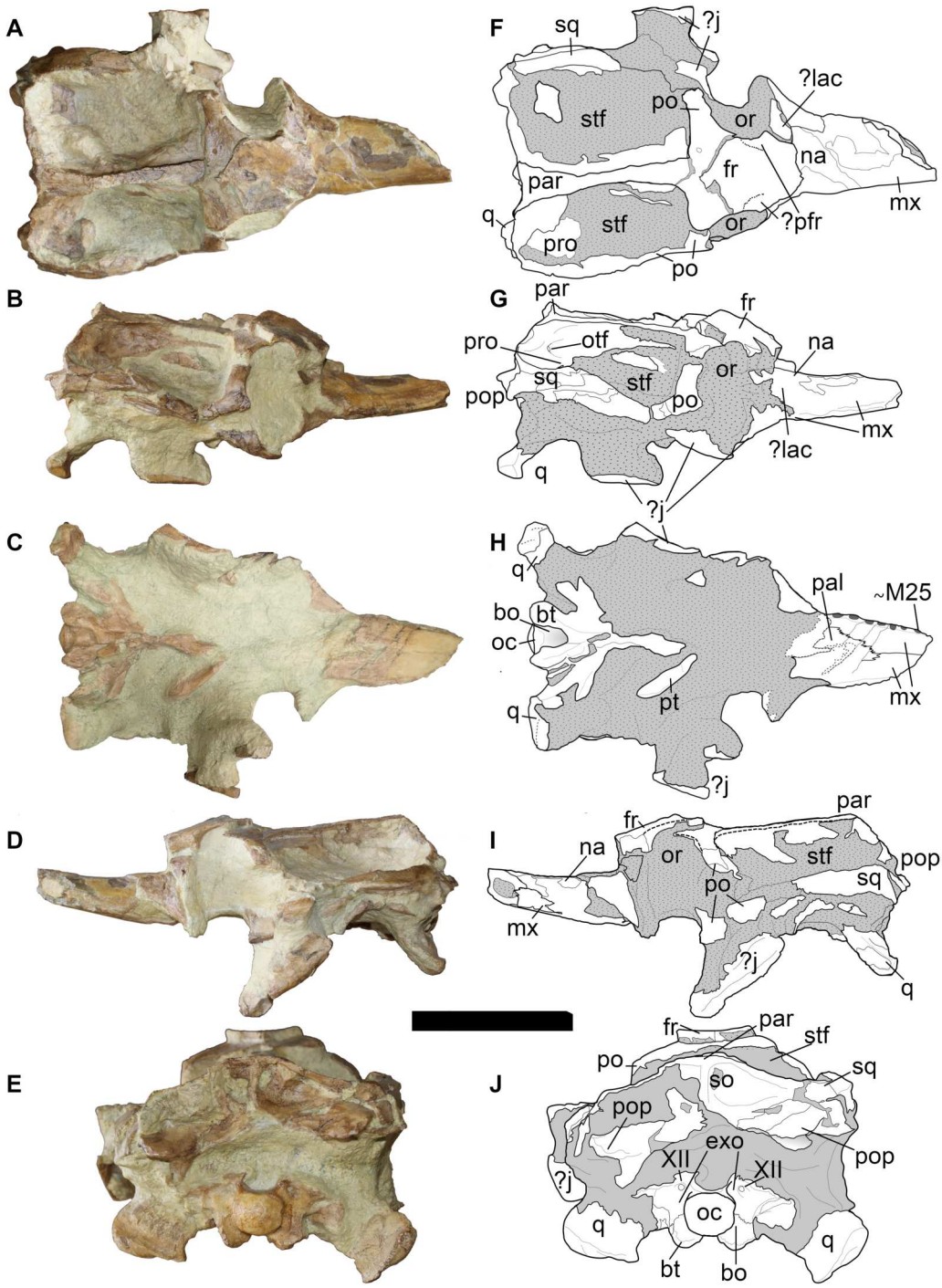

**Figure 5** **LPP specimen, orbital and posterior skull of referred specimen of *Bathysuchus megarhinus* gen. et. sp. nov. from the Upper Kimmeridgian of Franculés, Quercy, France, and interpretative drawings.** A, F, skull in dorsal view. B, G, skull in right lateral view. C, H, skull in ventral view. D, I, skull in right lateral view. E, J, skull in posterior view. Scale bar equals 10 cm.

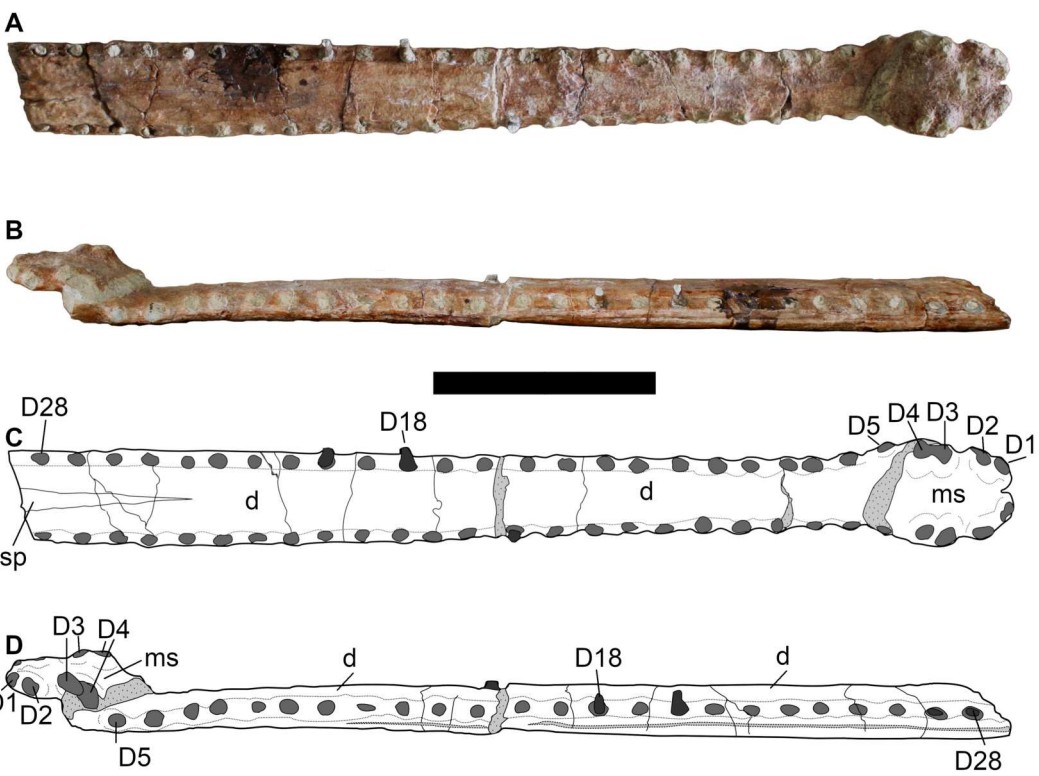

**Figure 6** **LPP specimen, mandibular symphysis of referred specimen of *Bathysuchus megarhinus* gen. et. sp. nov. from the Upper Kimmeridgian of Franculés, Quercy, France, and interpretative drawings.** A, C, mandible in dorsal view. B, D, mandible in left lateral view. Scale bar equals 10 cm.

*nasutus Andrews, 1913*; 'Steneosaurus' jugleri, and probably *Aeolodon priscus*); in dorsal view the external nares have an '8' shape, created by one enlarged anteriorly-directed and one dorsally-directed projections of the of the premaxilla (shared with *Mycterosuchus nasutus;* and possibly 'Steneosaurus' jugleri, Teleosaurus cadomensis (see *Eudes-Deslongchamps, 1870*; *Brignon, 2014* and 'Steneosaurus' megistorhynchus Geoffroy, 1831 (emend. *Eudes-Deslongchamps, 1866*)); the external nares are antero-dorsally oriented (shared with 'Steneosaurus' brevior (*Tate & Blake, 1876*), *Mycterosuchus nasutus*, *Platysuchus multiscrobiculatus* and the Chinese teleosauroid (IVPP V 10098, previously referred to as *Peipehsuchus* (see *Li, 1993*); reduced anteroposterior length of the external nares: more than 67% of the premaxillae total length is posterior to the external nares [shared with 'Steneosaurus' gracilirostris *Westphal, 1961* and the Chinese teleosauroid (IVPP V 10098), 'Steneosaurus' brevior, and *Lemmysuchus obtusidens* (*Johnson et al., 2017*); considerably pronounced lateral expansion of the premaxilla*; the anterior and anterolateral margins of the premaxillae are strongly anteroventrally deflected and extend ventrally (shared with 'Steneosaurus' brevior, *Mycterosuchus nasutus, Platysuchus multiscrobiculatus* and the Chinese teleosauroid) [note that the extent of the premaxillary deflection could be hidden in dorsoventrally compressed specimens. In order to avoid overinterpretations, in this study we exclusively discussed (and scored) this character in first-hand examined,
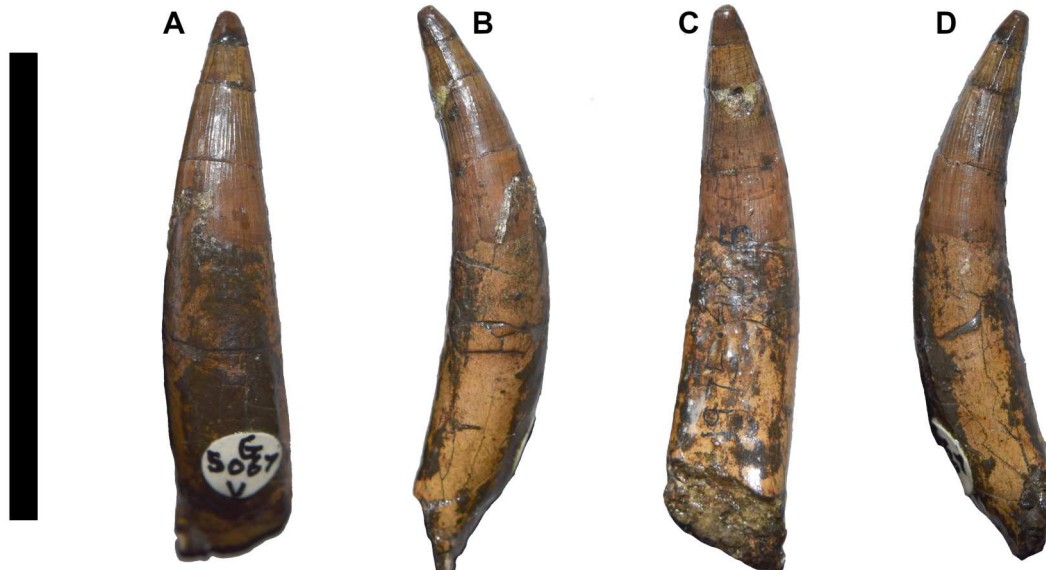

**Figure 7** DORCM G.05067iv tooth of *Bathysuchus megarhinus* gen. et. sp. nov. A, tooth in labial view. B, tooth medial-mesial view. C, tooth in lingual view. D, tooth in mesial-medial view. Scale bar equals 3 cm.

or obviously well-preserved specimens]; inconspicuously ornamented maxillary dorsal surface (shared with the Chinese teleosauroid, and *Aeolodon priscus*), consisting of a shallow irregular pattern of ridges and anastomosing grooves; nasal, prefrontal, lacrimal are also inconspicuously ornamented; absent/extremely reduced frontal ornamentation (shared with *Aeolodon priscus*); the rostrum narrows markedly immediately anterior to the orbits (shared with *Teleosaurus cadomensis* and *Mycterosuchus nasutus*); in dorsal view, the minimum interorbital width across the frontal is broader than the orbital width (shared with 'Steneosaurus' bollensis von *Jäger, 1828*, *Platysuchus multiscrobiculatus*, *Teleosaurus cadomensis*, 'Steneosaurus' brevior, and 'Steneosaurus' gracilirostris); small and reduced occipital tuberosities; anterior maxillary interalveolar spacing is sub-equal to longer than adjacent alveoli; lack of apical tooth ornamentation*; in the mandible, the fifth dentary alveolar pair is posterolaterally oriented and on the posterior end of the mandibular spatula (rather than posterior to the mandibular spatula)*; the ornamental pits on the dorsal osteoderms are circular and regularly organised in alternate rows (shared with *Aeolodon priscus*).

**Remarks**. The LPP specimen has a basicranial length of approximately 78 cm, which using the body length equations of *Young et al. (2016)* yields a 4 m total body length estimate. Based on the skull proportions of the most complete specimen (the LPP specimen), the holotype *Bathysuchus megarhinus* individual (NHMUK PV OR 43086) would have had a basicranial length of approximately 85 cm, and using the body length equations of *Young et al. (2016)* is estimated at 435 cm in body length. Comparing overlapping elements amongst
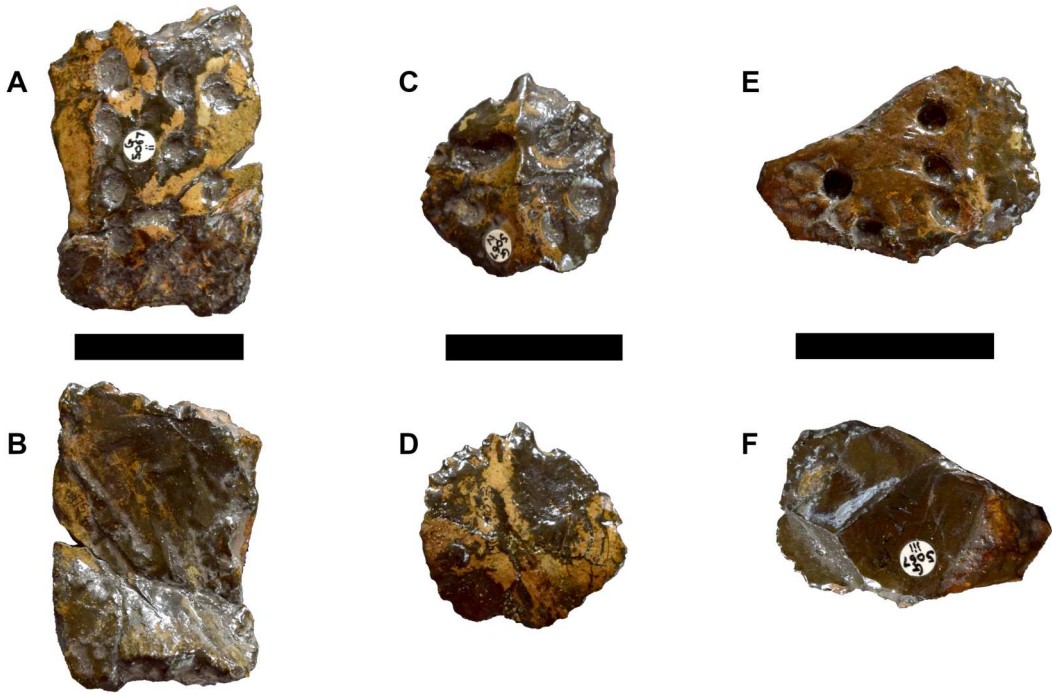

**Figure 8** DORCM G.05067ii-iv osteoderms of *Bathysuchus megarhinus* gen. et. sp. nov. A, Dorsal-sacral osteoderm DORCM G.05067ii in dorsal view. B, dorsal-sacral osteoderm DORCM G.05067ii in ventral view. C, caudal osteoderm DORCM G.05067iv in dorsal view. D, caudal osteoderm DORCM G.05067iv in ventral view. E, ventral osteoderm DORCM G.05067iii in view. F, ventral osteoderm DORCM G.05067iv in dorsal view. Scale bar equals 3 cm.

the available specimens, we notice that the DORCM specimen was undoubtedly larger than the holotype, but unfortunately its size cannot be confidently estimated.

**Geographical and stratigraphic range**. Kimmeridgian of England (*A. autissiodorensis* ammonite Zone) and France (*A. eudoxus* ammonite Zone)

# DESCRIPTION
## Cranial elements
### Premaxillae

The premaxillae of *Bathysuchus megarhinus* are ladle-shaped elements with a strongly convex dorsal side and strongly concave ventral surface (Figs. 2–4). The premaxillae of the holotype (NHMUK PV OR 43086) each bear five alveoli (although the last one is difficult to see in ventral view due to poor preservation). Unfortunately, the P5 alveoli cannot be seen in DORCM G.05067i due to the poor preservation of both the premaxilla posterior to the P4 alveoli and the premaxilla-maxilla suture (Fig. 3). Such a high alveolar count is unusual amongst teleosauroids, being known only in *B. megarhinus* and *P. multiscrobiculatus* (MNHNL TU895, SMNS 9930), as well as having been reported in *T. cadomensis* (see *Lamouroux, 1820*; *Eudes-Deslongchamps, 1869*; *Westphal, 1961*; *Westphal, 1962*; *Johnson et al., 2018*), 'Steneosaurus' deslongchampsianus (*Lennier, 1887*; *Godefroit, Vignaud &*

*Lieger, 1995*) new specimens referred to '*Steneosaurus*' *jugleri* (*Von Meyer, 1845*; *Schaefer, Püntener & Billon-Bruyat, 2018*), but not *Aeolodon priscus* (contra *Godefroit, Vignaud & Lieger, 1995*). The distribution of the premaxillary alveoli differs in teleosauroids and has phylogenetic importance. In *Bathysuchus, Aeolodon* (MNHN.F.CNJ 78), *Mycterosuchus* (CAMSM J.1420), and '*Steneosaurus*' *jugleri* (SCR011-406), in fact, the P1 and P2 alveoli are regularly circular/subcircular and laterally aligned. This differs in '*Steneosaurus*' *leedsi*, '*S.*' *heberti*, and Machimosaurini, where P2 is posterior and lateral compared to P1, giving the anterior premaxilla a tapering shape. In these taxa, the P1 and P2 alveoli are very close together, have an irregular (oval/teardrop shape) and in *Lemmysuchus obtusidens* they are separated by only a thin lamina (*Johnson et al., 2017*). In *Bathysuchus,* [and to a minor extent in *Aeolodon* (MNHN.F.CNJ 78), *Mycterosuchus* (NHMUK PV R 2617, CAMSM J.1420) and '*Steneosaurus*' *jugleri* (SCR011-406)], the lateral margins of the premaxillae are strongly laterally expanded, so that the P3-P4 alveoli are anteroposteriorly aligned on a more lateral plane than the external margin of the P2 alveoli. Posterolateral to the P2 alveolus is a noticeable diastema (separating it from the P3 alveolus). The P3 and P4 alveoli are also well separated, and the P5 alveolus is small, positioned dorsally compared to P1-4 and laterally and posteriorly oriented. This morphology is unique in *Bathysuchus megarhinus* and is clearly visible on NHMUK PV OR 43086 and the LPP (Quercy) specimen (Figs. 2–4 and 9).

The external nares are well preserved in both NHMUK PV OR 43086 and DORCM G.05067i, but less so in the LPP specimen, where only the anterior portion is partially exposed from the matrix that fills the narial cavity. The anteromedial and posteromedial margins of the external nares are exceptionally bulbous, and project anteriorly and dorsally, respectively (*Hulke, 1871*) (Figs. 2F, 2L, 3E, 3J and 9). This gives the external nares a peculiar '8-shape' in dorsal and anterior views (Figs. 2F, 2L, 3E, 3J and 9), which is not evident in any other teleosauroid besides *Mycterosuchus nasutus* (CAMSM J.1420, DF, *pers. obs.*). However, this area is too damaged in *Aeolodon* (MNHN.F.CNJ 78) to be confidently assessed (Figs. 9D–9E and 10). Overall, the external nares constitute a small length of the entire premaxillae: the portion of the premaxilla posterior to the external nares is more than 67% of its entire length, longer than in *A. priscus* (MNHN.F.CNJ 78), where it is approximately 60–65% (it is between 50–65% in the basal '*S.*' *gracilirostris* and '*S.*' *leedsi*). The anterior and posterior medial margins of the external nares are made by two bulbous projections of the premaxillae in the dorsal and anterior directions, respectively (*Hulke, 1871*) (Figs. 2F, 2L, 3E, 3J and 9D–9F). Faint ridges ornament the anterior margin of the anterior projection and the external nares of *B. megarhinus* (NHMUK PV OR 43086, DORCM G.050671i); these ridges are commonly present in other teleosauroids (e.g., *Lemmysuchus obtusidens* NHMUK PV R 3168).

The premaxillae in both Dorset specimens (NHMUK PV OR 43086 and DORCM G.05067i) are laterally expanded (such that they extend considerably more laterally than the anterior maxilla lateral margins) in line with the P3-P4 alveoli, and are also strongly ventrally deflected (Figs. 2–4 and 9F). The lateral expansion of the premaxillae cause P3-P4 to be positioned on a plane lateral to the rest of the premaxillary and maxillary alveoli. Due to the ventral deflection, the P1-P3 alveolar margin is also situated on a ventral

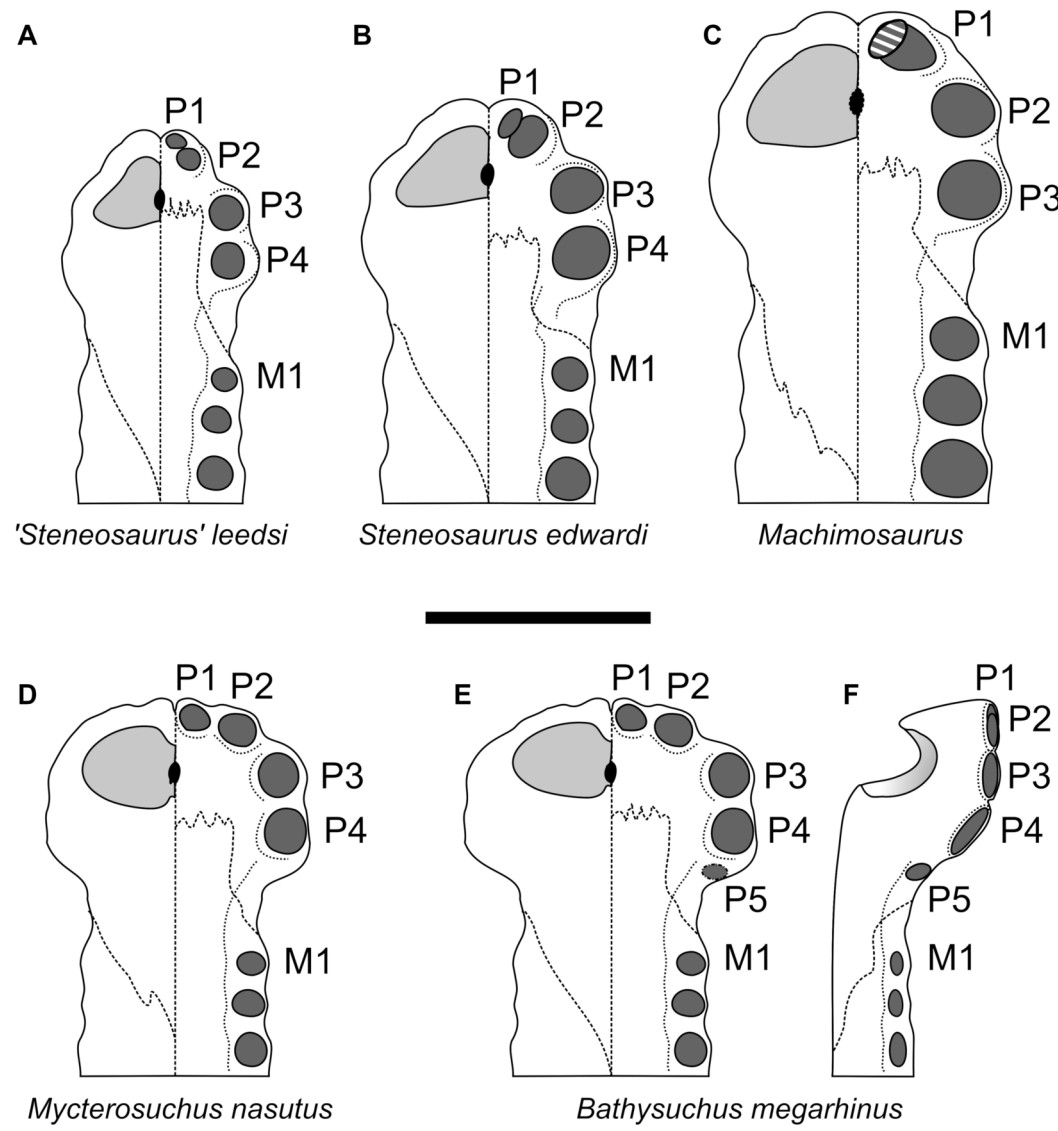

**Figure 9 Comparative plate of selected teleosauroids premaxillae.** The left side of each diagram depicts the dorsal view, the right side the palatal view. Scale bar equals 5 cm.

plane compared to the remaining snout dentition, including the P4 alveoli (which is more dorsally and slightly posteriorly oriented) (Figs. 2C, 2I, 3C, 3H, 4C, 4G and 9). The lateral premaxillary expansion is not uncommon in teleosauroids within the subclade including 'Steneosaurus' brevior and *Mycterosuchus nasutus* (Fig. 9). Yet, it is noticeably more extreme in *Bathysuchus megarhinus* than in the other taxa. It is possible that this condition has been slightly exaggerated by the preservation of the specimens. However, we see no reason why a dorsoventral compaction would result in moving the premaxillae downwards rather than buckling it on the same level as the maxillae (as it can be seen in *Mycterosuchus nasutus*, 'Steneosaurus' leedsi and other dorsoventrally flattened specimens of the Oxford Clay Formation) (Fig. 9). Notably the difference between the latter two taxa is still visible

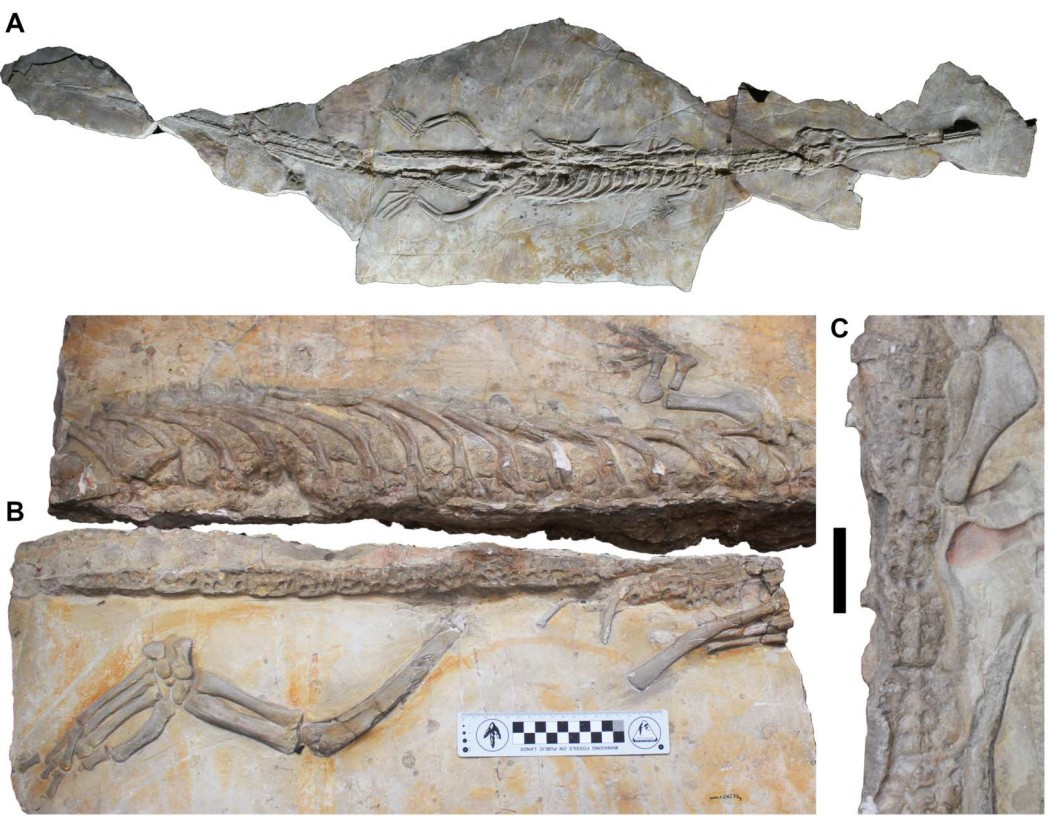

**Figure 10** *Aeolodon priscus* **MNHN.F.CNJ 78.** A, complete skeleton in dorsal view. B, dorsal region in dorsal view with details of the left forelimb and left hindlimb. C, details of the right forelimb. Scale bar in C equals 5 cm.

regardless of diagenetic flattening. Perhaps more convincingly, NHMUK PV OR 43086 and DORCM G.05067i are not significantly dorsoventrally distorted, as can be assessed by the well-preserved oval shape of their rostrum cross-section (Figs. 2–3). Finally, this set of features (that also affect the orientation and shape of the external nares) is present in the well preserved and undistorted LPP specimen, which is the strongest evidence that they are genuine (Fig. 4).

As reported by *Hulke (1871)*, the P4 alveoli are the largest alveoli in the premaxillae, and the P1 and P5 alveoli are the smallest (Figs. 2–4). In dorsal view, the premaxillae contact the maxillae via a slightly interdigitating, 'V-shaped' suture that reaches level to the M3 alveoli, or slightly posterior. In ventral view, the same suture has a straight anterior margin, creating a sub-square profile with the anterior-most side reaching in between the P3 and P4 alveoli (Figs. 2C, 2I, 3C, 3H, 4C and 4G). The ornamentation of the dorsal surface of the premaxillae is weak and shallow, as in *A. priscus* (MNHN.F.CNJ 78), and considerably less pronounced than in *M. nasutus* (NHMUK PV R 3577, and CAMSM J.1420), 'S.' *brevior* (NHMUK PV OR 14781) and taxa within Machimosaurini (e.g., *Lemmysuchus obtusidens* NHMUK PV R 3168; *Machimosaurus buffetauti* V1600Bo) (Figs. 2–4 and 11).

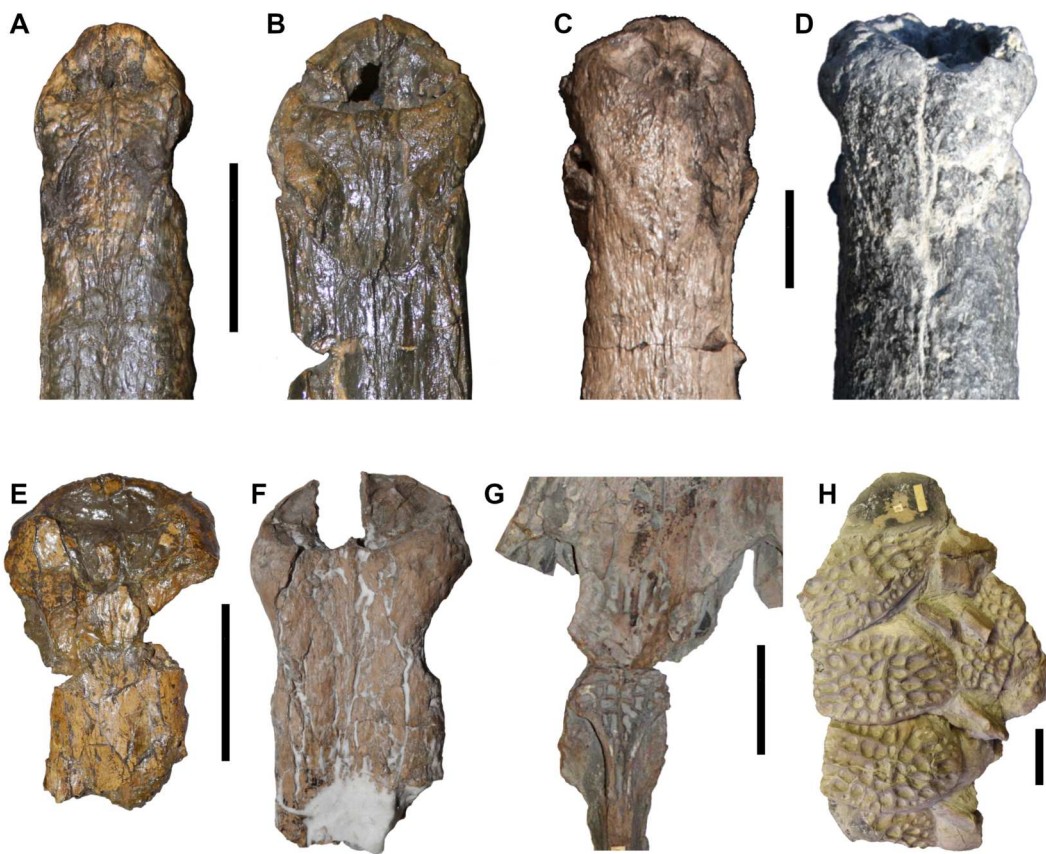

**Figure 11** **Comparative plate of selected teleosauroids premaxillae showing different degrees of skull and postcranial ornamentation.** A, *'Steneosaurus' leedsi* NHMUK PV R 3806. B, *Steneosaurus edwardsi* NHMUK PV R 3701. C, *Lemmysuchus obtusidens* NHMUK PV R 3168. D, *Machimosaurus buffetauti* MPV V1600.Bo. E, *Bathysuchus megarhinus* DORCM G.05067i. F, G, H, *Mycterosuchus nasutus* CAMSM J.1420; F, premaxilla; G, orbital area; H, dorsal osteoderms. Scale bars equal 5 cm.

As well as NHMUK PV OR 43086 and DORCM G.05067i, the LPP specimen shows the combination of characteristic premaxillary and maxillary features of *Bathysuchus*: (1) five premaxillary alveoli (with the P5 being small and posteriorly oriented); (2) extreme lateral expansion of the premaxillae; (3) the premaxilla posterior to the external nares is over 67% of its entire length; (4) the P1 and P2 alveoli are laterally aligned and do not form a couplet; (5) the P3-P4 alveoli are anteroposteriorly aligned more laterally than the P1-P2; (6) P1-P3 alveoli are situated on a ventral plane compared to the rest of the premaxillary alveoli; (7) faint, shallow ornamentation on the premaxillae. As the interpremaxillary septum is slightly broken/covered by matrix in the LPP specimen, we cannot assess whether the external nares were '8-shaped' with bulbous anteromedial and posteromedial margins. They were, nevertheless, anterior-dorsally oriented and shorter than their width.

### Maxillae

The maxillae are partially preserved in both NHMUK PV OR 43086 and DORCM G.05067i (although they are slightly more complete in NHMUK PV OR 43086), but their sutures

with adjacent posterior elements cannot be assessed (Figs. 2–4) (however they are more completely preserved in the LPP specimen, see below). The maxillae form a substantial part of the rostrum, with sub-parallel lateral margins in dorsal view. The rostrum is dorsoventrally flattened (oval in cross section with a horizontal long axis) (Figs. 2E and 2K). The dorsal and lateral surfaces of NHMUK PV OR 43086 and DORCM G.05067i are weakly ornamented with a shallow network of ridges, rugosities and anastomosing grooves (Figs. 2–3 and 11). The density and depth of the maxilla ornamentation varies in teleosauroids, and in semi-aquatic taxa the loss of dermatocranial and osteoderm ornamentation has been linked to a pelagic lifestyle (*Young et al., 2013*; *Clarac et al., 2017*) (see Discussion). This is similar to the evolutionary trend in metriorhynchids, which plesiomorphically had well ornamented dermatocrania that independently became 'smoother' in numerous lineages through time (*Young et al., 2013*).

With reference to the LPP specimen (Figs. 4–5) (also see Fig. 2 *Vignaud et al., 1993*, and Plate 12, *Vignaud, 1995*), the rostrum of *B. megarhinus* makes up approximately 71% of the total basicranial length, compared with ∼73% in *M. nasutus* (NHMUK PV R 3577, CAMSM J.1420) and *A. priscus* (MNHN.F.CNJ 78), and ∼74% in *P. multiscrobiculatus* (SMNS 9930). The rostral length in teleosaurids ranges from ∼55–75%, with *Machimosaurus mosae Sauvage & Liénard, 1879* having the lowest rostrum/basicranial length ratio in teleosaurids and '*S.' leedsi* (∼72%), *M. nasutus*, '*Steneosaurus' deslongchampsianus*, and *A. priscus* the highest (∼74%). This suggests that rostrum length is a plastic feature in teleosauroids, with different clades independently diverging from the plesiomorphic longirostry towards extreme longirostrine or mesorostrine/brevirostrine forms.

Given the incomplete preservation of these specimens, it is impossible to provide a precise tooth count for *B. megarhinus*. The Quercy LPP specimen (*Vignaud et al., 1993*; *Vignaud, 1995*) has 28–30 preserved maxillary alveoli, but the tooth row is posteriorly incomplete. Approximately 24 anterior-most alveoli are preserved on each side of NHMUK PV OR 43086, and only the three anterior-most alveoli are preserved on the right maxilla of DORCM G.05067i. The maxillary interalveolar spacing is regular and in the anterior third of the maxilla, it is longer than the adjacent alveolar length (Figs. 2–4). The palatines are not visible in any of the Dorset specimens. *Vignaud et al. (1993)* reported that in the LPP specimen the palatines occupy the entire width of the beginning of the rostrum (in line with the anterior margin of the orbits), and the maxillae are reduced to thin lateral bands. It is not possible to accurately estimate the anterior extent of the palatine-maxillae suture in relation to the tooth count, but it certainly does not extend more than approximately ∼5 cm anterior to the posterior end of the rostrum, roughly corresponding to the position of the M27-M30 alveoli (also see Fig. 2 (*Vignaud et al., 1993*); Plate 12, (*Vignaud, 1995*)).

The maxillae of the LPP specimen are anteroposteriorly elongated with sub-parallel lateral margins as in other teleosauroids (e.g., *Lemmysuchus obtusidens* NHMUK PV R 3168). In dorsal view, the rostrum narrows markedly immediately anterior to the orbits, which is also seen in *T. cadomensis* (MNHN.F AC 8746), *A. priscus* (MNHN.F.CNJ 78) and *M. nasutus* (NHMUK PV R 3577). The LPP specimen has approximately 28–30 preserved maxillary alveoli, although the posterior-most alveoli are not preserved, and M30 does not correspond with the end of the tooth row.
### Nasals, frontal and postorbital

The nasals are elongated and triangularly shaped as in other thalattosuchians (*Andrews, 1913*; *Young & Steel, 2014*) and do not contact the premaxilla, although it is impossible to precisely determine their anterior extent. Based on the break of the rostrum (M25-M26) and the posterior skull, it can be estimated to be around the M27-M30 interval. The frontal (although the anterior part is severely distorted) is a large, single bone that is similar in shape to that in most other teleosauroids (e.g., 'S.' *leedsi* NHMUK PV R 3806; see *Andrews, 1909*; *Andrews, 1913*). The minimum interorbital width across the frontal is broader than the orbital width, shared with 'S.' *bollensis* (SMNS 51753), *P. multiscrobiculatus* (SMNS 9930), *T. cadomensis* (MNHN.F AC 8746), 'S.' *brevior* (NHMUK PV OR 14781) and 'S.' *gracilirostris* (NHMUK PV OR 14792). One peculiar feature is the absent/faint ornamentation on the frontal, which is also seen in *Aeolodon priscus* (note that *Vignaud et al., 1993* suggested that this feature could be a juvenile characteristic) (see 'Discussion') (Figs. 5 and 10). Only the right postorbital is near-completely preserved (Fig. 5). However, the posterior end is broken and slightly anteroventrally displaced. The postorbital forms the lateral and posteroventral borders of the supratemporal fenestra. The frontal and postorbital form the postorbital bar (although the contact between the two bones cannot be seen clearly), which is mediolaterally short and slightly anteroposteriorly slender.

### Postorbital cranium

The parietal is large, fused and unornamented (Fig. 5). The supratemporal fenestrae are large, anteroposteriorly elongated and sub-rectangular in shape (in dorsal view), similar to those seen in most teleosauroids [e.g., 'S.' *leedsi* (NHMUK PV R 3806), *A. priscus* (MNHN.F.CNJ 78), *M. nasutus* (NHMUK PV R 3577) and *S. bollensis* (e.g., SMNS 51753)]. Their lateral margins appear relatively straight in dorsal view (Figs. 5A and 5F) and concave in lateral view (although the right supratemporal fenestra is diagenetically disfigured in lateral view) (Figs. 5B, 5G, 5D and 5I). The squamosals (Fig. 5) are damaged, especially in the anterior area. However, they appear to be L-shaped, as in other teleosauroids (e.g., 'S.' *leedsi* NHMUK PV R 3806). The squamosal forms the posterolateral border of the supratemporal fenestra.

### Occiput

In occipital view (Figs. 5E and 5J), the supraoccipital is dorsoventrally tall. The exoccipitals (which make up the majority of the occiput) are large, tilted dorsally and slightly concave and flared dorsoventrally. The cranial nerve XII foramen is level with the foramen magnum, and the quadrates appear relatively small, each bearing two separate hemicondyles. The occipital tuberosities are small and reduced as in other teleosauroids such as 'S.' *leedsi* (NHMUK PV R 3806) (one exception is '*Steneosaurus*' *heberti* Morel de Glasville, 1876 (MNHN.F 1890-13), in which they are large and bulbous). The exoccipitals meet at the midline and are dorsoventrally broad, relatively anteroposteriorly short and extend horizontally (Figs. 5E and 5J). The paraoccipital process is the same size as the remainder of the exoccipital. In the LPP specimen, the left exoccipital is ventrally displaced.

### Mandible

The LPP mandible is preserved up to the D28 alveolar pair (Fig. 6). The spatulate anterior area is broken and its anterior part has been dorsally displaced. The anterior spatula is similar in shape and proportion to that of other longirostrine teleosaurids: D1-D2 are widely separated by a noticeable notch (presumably for the P3 tooth) from D3-D4. D4 and D5 are also separated by a diastema. However, uniquely in *Bathysuchus megarhinus*, the D4-D5 interalveolar space is reduced and D5 sits on the posterior end of the "spatula". Consequently, D5 is on the same lateral plane as D2-3-4 rather than being in line with the other dentary teeth of the symphyseal area. Posterior to the spatulate area, the dentary interalveolar spacing is consistently sub-equal to one alveolar length (and slightly increases along the tooth row in posterior direction) (Fig. 6).

### Dentition

No teeth are preserved in the type specimen, with DORCM G.05067v having one loose tooth crown that is well preserved enough to allow description (Fig. 7). The crown is small, only ~17mm in apicobasal length, with a high crown base average diameter/ crown height ratio (~2.5). The crown is sub-circular in cross section at its base, slightly laterally compressed in the apical third, and it is weakly curved in medial direction. The enamel is finely ornamented by continuous parallel apicobasally aligned ridges that are densely packed and low-relief. The ridges do not reach the apex of the tooth, but stop two-thirds up the crowns in both the mature and unerupted teeth (left P4 and M2). DORCM G.05067v has one visible carina on both mesial and distal margins. The carinae are easier to detect on the apical third of the crown, and they are smooth, as no denticles can be observed (even using optical aids). Following the functional classification of Mesozoic marine reptile teeth by *Foffa, Young & Brusatte (2018)*, *B. megarhinus* (DORCM G.050761v) falls in the 'Pierce' guild, along with the other non-Machimosaurini teleosauroids in the dataset.

There is one tooth (P4) preserved *in situ* in the LPP skull, and three more in the mandible (D15 on the right side, D18 and D20 on the left side) (Fig. 6). The teeth are slender, sub-circular in cross section, apicobasally elongated and weakly curved in the lingual direction. Unfortunately, they are all incomplete and partially covered with matrix, hiding their apices and enamel ornamentation.

### Osteoderms

No osteoderm was found associated with the holotype or LPP specimen, but three osteoderms (two dorsal-sacral and one ventral) are preserved in DORCM G.05067i-iv (Fig. 8). Based on their respective rectangular and sub-circular shapes, the dorsal osteoderms presumably come from one of the paramedian series of the dorsal series (DORCM G.05067ii) (Figs. 8A–8B) and the anterior tail (DORCM G.05067iv) (Figs. 8C–8D). All osteoderms are relatively reduced in size and thickness compared to all other teleosaurids except *Aeolodon priscus* They are ornamented with small circular/sub-circular pits that are organised in alternate rows, unlike in the usual 'starburst' patterns of tear-drop/irregular shaped pits of most teleosauroids (e.g., machimosaurins; see *Young & Steel, 2014* and *Johnson et al., 2017*). The regular shape and arrangement of these pits is similar to the morphology observed in *A. priscus* (MNHN.F.CNJ 78) (Fig. 10) and '*Steneosaurus*'

*jugleri* (*Godefroit, Vignaud & Lieger, 1995*). The caudal osteoderm, DORCM G.05067iv, has a well-developed medial keel that is not present in the other two osteoderms (although the keel may be missing due to preservation) (Figs. 8A and 8C). The third osteoderm, DORCM G.05067iii, differs somewhat from the others due to the ornamental pits being more widely separated from one another, with a flat external surface (Figs. 8E–8F). Overall, the osteoderms of *B. megarhinus* are poorly ornamented compared to other teleosauroids (e.g., see *Andrews, 1913*; *Johnson et al., 2017*), a character shared with *A. priscus* (MNHN.F.CNJ 78) (Fig. 8, 10B, 10C and 11H), and 'Steneosaurus' *jugleri* (SCR010-312) (*Godefroit, Vignaud & Lieger, 1995*).

Finally, an osteoderm NHMUK PV OR 40105 of similar shape, preservation and ornamentation to DORCM G.05067ii was found associated with the matrix of *Plesiosuchus manselii* (NHMUK PV OR 40103), which was from the *A. autissodorensins* Sub-Boreal ammonite Zone (Lower Kimmeridge Clay Formation of Kimmeridge Bay)—the same locality of the UK specimens of Bathysuchus megarhinus. However, given its poor preservation, we cannot be certain it belongs to *Bathysuchus megarhinus*.

## PHYLOGENETIC ANALYSIS

### Methods

We conducted a phylogenetic analysis to test the evolutionary relationships of *Bathysuchus megarhinus* gen. nov. within Thalattosuchia, using a modified version of the dataset published by *Ösi et al. (2018)*, which is continuously being updated, as it forms the foundation of the ongoing Crocodylomorph SuperMatrix Project. The dataset was first presented in *Ristevski et al. (2018)*; however, it has been extensively updated subsequently (see *Ösi et al., 2018* for full details). All data are summarised in Supplementary data files.

The current dataset consists of 140 crocodylomorph OTUs (70 of which are thalattosuchians, including 18 teleosauroids, seven basal metriorhynchoids and 42 metriorhynchids) scored for 456 characters. Of these 456 characters, 25 characters representing morphoclines were treated as ordered (see Data S1) and *Postosuchus kirkpatricki* (*Chatterjee, 1985*) was used as the outgroup taxon. The differences between our analyses and those presented by *Ösi et al. (2018)* are: (1) the rescoring of *B. megarhinus*, *M. nasutus* and *A. priscus*; (2) the rescoring of the Chinese teleosauroid (IVPP V 10098) OTU; and (3) a re-organisation of the character list, with the addition of 3 new characters (Ch. 31, 274, 275), deletion of one (former Ch. 42), and inclusion of two new anatomical sections (palaeoneuroanatomy and craniomandibular pneumaticity). The character scoring for *B. megarhinus* was based on first-hand examination of the holotype by DF, MMJ and MTY, as well as first-hand examination of the referred DORCM specimen *Bathysuchus* by DF and the LPP specimen by MMJ. Due to the poor preservation and incompleteness of these specimens, *B. megarhinus* was scored for 159 out of 456 characters (35.0%).

The protocol used to analyse the dataset is the same adopted by *Ösi et al. (2018)*, and it is described in detail in Data S1.

**Figure 12  Results of the phylogenetic analysis.** Simplified strict consensus trees of the 85 most parsimonious cladograms of Teleosauroidea within Crocodylomorpha.

# RESULTS

The phylogenetic analysis produced 85 most parsimonious trees (MPTs) with 1,494 steps (ensemble consistency index (CI) = 0.414; ensemble retention index (RI) = 0.841; rescaled consistency index (RCI) = 0.348; ensemble homoplasy index (HI) = 0.586) (Fig. 12). The overall strict consensus topology recovered from this analysis is slightly different from those presented by *Ristevski et al. (2018)* and *Ösi et al. (2018)*.

The overall picture of crocodylomorph interrelationships are similar to those found in previous iterations of this merged dataset (*Ristevski et al., 2018*; *Ösi et al., 2018*). Thalattosuchia is monophyletic within Crocodyliformes but outside Metasuchia. Within Thalattosuchia, both Teleosauroidea and Metriorhynchoidea are recovered as monophyletic, with *Eopneumatosuchus colberti* (*Crompton & Smith, 1980*) from the Kayenta Formation of Arizona, USA (*Curtis & Padian, 1999*) as their closest outgroup. In Metriorhynchoidea, *Pelagosaurus typus* (*Bronn, 1841*) is positioned as a basal metriorhynchoid, and Metriorhynchidae, Metriorhynchinae, Rhacheosaurini, Geosaurinae and Geosaurini are all monophyletic. Within Teleosauroidea, '*Steneosaurus gracilirostris*' is the basal-most species, and there are two large subclades here identified as 'clade T' (including poorly known genera and species (e.g., *Platysuchus*, *Bathysuchus*, *Teleosaurus* and *Mycterosuchus*) that are predominately long-snouted) ) and 'clade S' (including '*S.*' *leedsi*, the durophagous tribe Machimosaurini Machimosaurini and all taxa in between) (Fig. 12).

Within 'clade T', *Bathysuchus megarhinus* is found as sister taxon to *Aeolodon priscus* in a clade also containing *Mycterosuchus nasutus* (Fig. 12). This subclade is in turn closely

related to *Teleosaurus cadomensis*, *Platysuchus multiscrobiculatus*, '*Steneosaurus*' *brevior* and the Chinese teleosauroid (Fig. 12) within 'clade T'.

Note that our dataset does not include the recently catalogued new available materials of '*Steneosaurus*' *jugleri* from the Late Jurassic of Switzerland (*Schaefer, Püntener & Billon-Bruyat, 2018*). The description of the new material and assessment of the validity of this taxon exceeds the scope of the present manuscript. However, given its relative completeness (associated cranial and postcranial material), and the strong similarities (reduced cranial ornamentation and premaxillary alveolar count and arrangement) that this taxon shares with *Bathysuchus*, we predict that these specimens are bound to improve our knowledge of the relationships of the T-clade.

## DISCUSSION

### Comparisons with other teleosauroids

*Bathysuchus megarhinus* shares a number of characters with other teleosauroids, most notably with a handful of long-snouted taxa (e.g., *Mycterosuchus nasutus* and *Aeolodon priscus*, '*Steneosaurus*' *jugleri*). As mentioned in the description, the high premaxillary alveolar count (five) is unusual, and is only seen in *B. megarhinus* (NHMUK PV OR 43086, DORCM G.05067i, LPP specimen), *P. multiscrobiculatus* (MNHNL TU895, SMNS 9930 MMJ *pers. obs.*) and *T. cadomensis* (see *Lamouroux, 1820*; *Westphal, 1962*); '*Steneosaurus*' *deslongchampsianus* (*Godefroit, Vignaud & Lieger, 1995*; *Savalle, 1876*) and '*Steneosaurus*' *jugleri* (*Schaefer, Püntener & Billon-Bruyat, 2018*), but not *Aeolodon priscus* (contra *Godefroit, Vignaud & Lieger, 1995*). The peculiar premaxillary alveolar distribution of the P1-P2 and the P3-P4 alveoli are characteristic of both *B. megarhinus* (NHMUK PV OR 43086, LPP specimen, DORCM G.050671i) and *M. nasutus* (CAMSM J.1420 DF, *pers. obs.*), which is one of a number of features unique to these taxa (Fig. 9). Well-developed anterior premaxillary projections seen in *B. megarhinus* (NHMUK PV OR 43086 and DORCM G.050671i—but not in LPP specimen, as its narial cavities are infilled with matrix) are also present in *M. nasutus* (CAMSM J.1420, DF *pers. obs.*) (Figs. 2–3). This feature is considerably less clear in other teleosauroids, including closely related taxa such as *Aeolodon priscus* (MNHN.F.CNJ 78), but also '*Steneosaurus*' *leedsi* (NHMUK PV R 3806) and the Chinese teleosauroid (IVPP V 10098). *Bathysuchus megarhinus* shares three premaxillary features in with *P. multiscrobiculatus* (SMNS 9930, MNHNL TU895), '*Steneosaurus*' *brevior* (NHMUK PV OR 14781), *Mycterosuchus nasutus* (NHMUK PV R 2617; CAMSM J.1420), the Chinese teleosauroid (IVPP V 10098), and '*Steneosaurus*' *jugleri*.

(1)    The anterolateral margins of the premaxillae are strongly anteroventrally deflected (Figs. 2–4 and 9). In other teleosauroids (e.g., '*S.*' *leedsi* NHMUK PV R 3806) the anterior and anterolateral premaxillary margins are either not anteroventrally deflected, or not as strongly;

(2)    The anterior external nares face anteriorly (or anterodorsally) (Figs. 2–4 and 9). In other teleosauroids (e.g., '*S.*' *leedsi* NHMUK PV R 3806) the external nares face mainly dorsally;

(3)        The premaxilla is laterally expanded, in line with the P3-P4 alveoli.

In other teleosauroids (e.g., *'S.' leedsi* NHMUK PV R 3806, *S. edwardsi* NHMUK PV R 3701, Machimosaurini) (*Young & Steel, 2014*) the lateral expansion and the ventral deflection are not as clear, although we noticed that this feature may be present in *Mycterosuchus nasutus* (NHMUK PV R 3577; CAMSM J.1420), *Aeolodon priscus* (MNHN.F CNJ 78) and *'Steneosaurus' jugleri* (SCR011-406). Furthermore, in *Aeolodon* the premaxillae are laterally expanded and may be ventrally deflected, but both features may be concealed by the diagenetic deformation of the specimens (Fig. 9). Nevertheless, even allowing for deformation the extent of these features do not reach the extreme morphology present in every *B. megarhinus* specimen.

The teeth of *B. megarhinus* are unusual, as the enamel ridges do not continue onto the apical region (Fig. 7). This feature has not been observed in any other described teleosauroid, in which the enamel ridges are more densely packed and reach the apex (e.g., *M. nasutus* NHMUK PV R 3577, CAMSM J.1420; *A. priscus* MNHN.F CNJ 78; *'S.' bollensis* MNHNL TU799) (*Andrews, 1913*). However, it is worth noting that it has been observed in an undescribed MNHN teleosauroid (one tooth in association with a partial skull) and NHMW 1884 (from the 'Lias' of Germany), which is labelled as *'Teleosaurus'* (however, this tooth is laterally compressed, with discontinuous ridges that are more prominent than in *Bathysuchus*) (MMJ *pers. obs.*). The same feature is also visible in one tooth referred to *'Steneosaurus' jugleri* (TCH005-151) that is still wanting description (*Schaefer, Püntener & Billon-Bruyat, 2018*).

## Reduced ornamentation and possible pelagic adaptations in Teleosauroidea

The premaxillary and maxillary dorsal and lateral surfaces of *B. megarhinus* (NHMUK PV OR 43086, DORCM G.05067i) are particularly weakly ornamented, which is similar to the condition in the Chinese teleosauroid (IVPP V 10098), *A. priscus* (MNHN.F CNJ 78), and *'Steneosaurus' jugleri* (SCR011-406) (Figs. 5–8 and 10) (*Schaefer, Püntener & Billon-Bruyat, 2018*). This differs from most other teleosauroids, both closely related taxa and more distant relatives, that retain the plesiomorphic condition of strongly ornamented rostrum and skull roof (Fig. 11) (e.g., *M. nasutus* NHMUK PV R 3577, CAMSM J.1420; *'S.' brevior* NHMUK PV OR 14781; *L. obtusidens* NHMUK PV R 3168).

The shape and arrangement of osteoderm pits are similar in *B. megarhinus* (DORCM G.050671i), *A. priscus* (MNHN.F.CNJ 78) (Figs. 8, 10 and 11), and *'Steneosaurus' jugleri* (SCR010-312) (see *Schaefer, Püntener & Billon-Bruyat, 2018*). These pits are large, but few in number and well separated from one another when compared to other teleosauroids (Fig. 11). However, there is one notable difference between the osteoderms of *B. megarhinus* and *A. priscus*: the thoracic osteoderms of *B. megarhinus* are not keeled, whereas longitudinal keels are present on all osteoderms in *A. priscus* (MNHN.F.CNJ 78), even the cervical ones (Figs. 8 and 10).

The ornamentation of the dorsal-sacral osteoderms of *B. megarhinus* (DORCM G.05067i) radically differs from the irregular, reticular pattern seen in *M. nasutus* (NHMUK PV R 3577, CAMSM J.1420) (Fig. 11). The pronounced reduction in dermatocranial and

osteoderm ornamentation are characters shared between *B. megarhinus* and *A. priscus*, and are unique to these two species within Teleosauroidea (Figs. 8, 10 and 11). This contrast is striking considering that the heavily ornamented *Mycterosuchus nasutus* is sister taxon to *Aeolodon* + *Bathysuchus*.

The shift from highly ornamented dermal bone to low levels of ornamentation (or no ornamentation) is characteristic of the shift from amphibious to pelagic forms in Crocodylomorpha (*Clarac et al., 2017*). *Clarac et al. (2017)* outlined a possible mechanism for the increase in bone ornamentation in amphibious pseudosuchians, as a way to increase their basking efficiency. As dermatocranial and osteoderm ornamentation is highly vascularised, the overlying soft tissue can drive heat radiation to the bones. We can therefore hypothesise that there was a thermoregulatory shift in the lineage from *Mycterosuchus nasutus* to *Aeolodon* + *Bathysuchus*. Between the Callovian taxon *M. nasutus*, and the late Kimmeridgian-early Tithonian *Aeolodon* + *Bathysuchus* subclade there was a reduction in the size and thickness of their osteoderms, as well as a pronounced reduction in their dermatocranial and osteodermal ornamentation. We here hypothesise that this was in response to living in different habitats: heavily ornamented forms in semi-aquatic coastal environments and weakly ornamented forms in a more pelagic habitat, in which basking was less important (or perhaps impossible). A similar reduction in dermatocranial and osteodermal ornamentation is also seen in Metriorhynchidae, which are well established as fully pelagic thalattosuchians (*Fraas, 1902*; *Andrews, 1913*; *Fernández & Gasparini, 2008*; *Herrera, Fernández & Gasparini, 2013*; *Herrera et al., 2017*).

There is further evidence for a more pelagic lifestyle in the post-cranium of *A. priscus* (unfortunately the post-cranium of *B. megarhinus* is largely unknown). The largest *A. priscus* specimen, MNHN.F.CNJ 78, has proportionally very small dorsal osteoderms, proportionally short tibiae, and reduced forelimbs that have become more flipper-like. These reduced limb measurements contrast with almost all other teleosauroids with known postcranial skeletons (Fig. 13) (Data S2). To quantitatively demonstrate this, we selected two limb measurements to examine. The first is a ratio of the forelimb and hindlimb propodials (humerus: femur or H:F). This was chosen as a proxy for the relative reduction of the forelimb, as few metriorhynchid specimens have associated forelimb stylopodial elements, and even fewer a preserved manus. Within Metriorhynchidae the morphofunctional changes that transformed the forelimb into a hydrofoil-like structure resulted in it being greatly reduced in relative size compared to the hindlimb (e.g., *Fraas, 1902*). Secondly, we used the tibia: femur (T:F) ratio, because within Crocodylomorpha the tibia is proportionally reduced in size relative to the femur in numerous lagoonal and pelagic taxa (*Fraas, 1902*; *Andrews, 1913*; *Wu, Russell & Cumbaa, 2001*; *Schwarz, Frey & Martin, 2006*; *Buscalioni et al., 2011*). We produced and examined two distinct datasets including thalattosuchians, and extant crocodylomorphs (H:F—26 teleosauroids; 10 metriorhynchoids; 39 extant specimens; and T:F—31 teleosauroids; 11 metriorhynchoids; 39 extant specimens) (Data S2). The measurements of extant specimens were selected from *Iijima, Kubo & Kobayashi (2018)* dataset, while thalattosuchian measurements were personally taken by DF, MTY or MJ or come from *Mueller-Töwe (2006)* (Data S2).
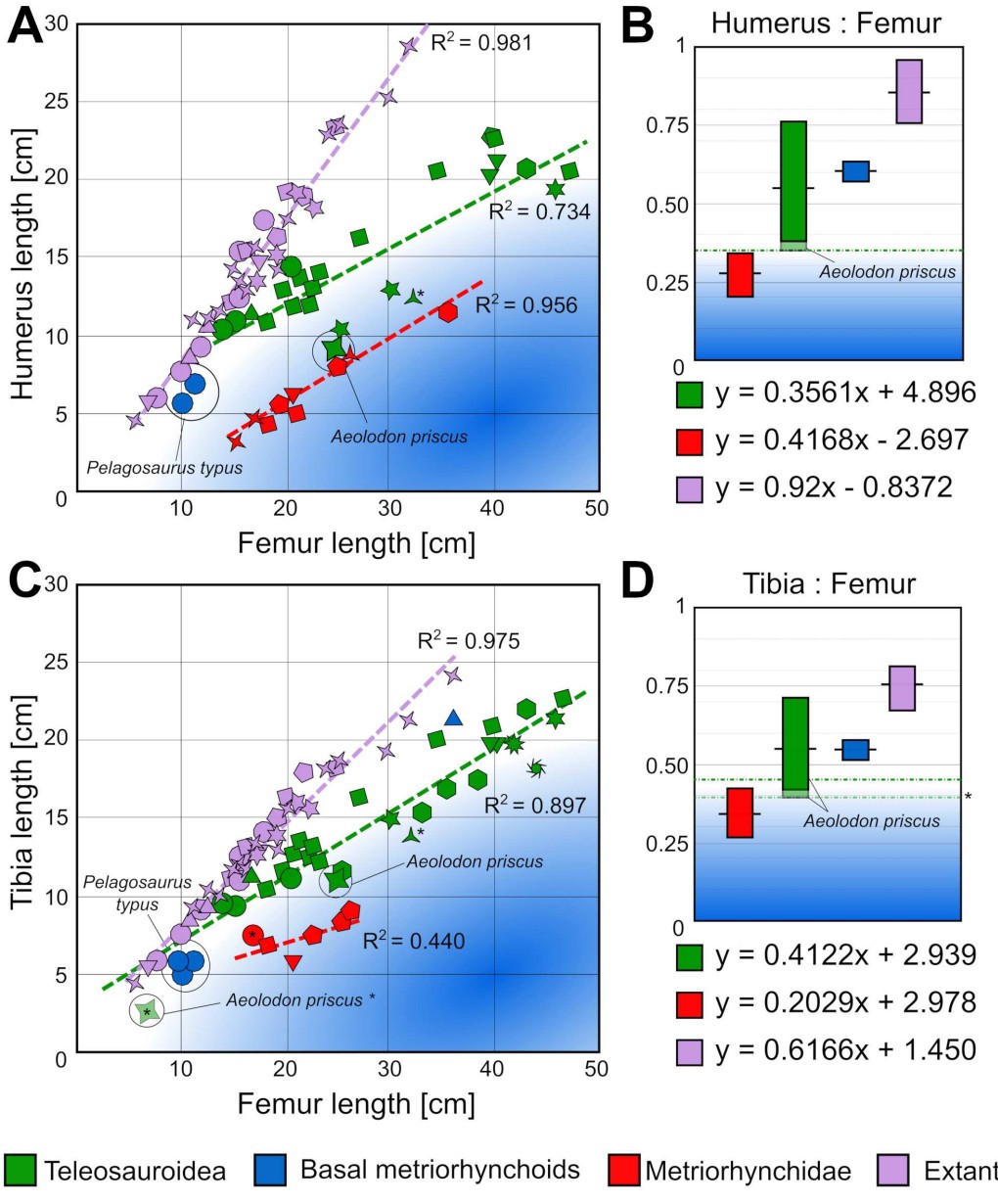

**Figure 13 Scatter plots showing the limb proportions of *Aeolodon priscus* compared to other teleosauroids, metriorhynchoids and extant crocodylians.** A, humerus length vs femur length scatterplot, and B, known ranges of humerus:femur ratio in thalattosuchian and extant crocodylomorphs. C, Tibia length vs femur length scatterplot, and D, known ranges of Tibia:femur ratio in thalattosuchian and extant crocodylomorphs. The humerus:femur and tibia:femur ratios of *Aeolodon priscus* approaches those of fully pelagic metriorhynchids, in the low end of the known range of teleosauroids.* Indicates a juvenile specimen; each symbol represents a genus.

The first intriguing feature is that the H:F and T:F linear equations of teleosauroids, metriorhynchids and extant species noticeably differ. This further highlights that the body-plans of these three groups where distinctly different, as the linear equations of basicranial length-total body length and femoral length-total body length are already known to

have noticeably differed between them (see *Young et al., 2011*; *Young et al., 2016*). Overall, relative to extant crocodylians, thalattosuchians had proportionally: longer skulls relative to total body length, shorter femoral lengths relative to total body length, shorter humeral lengths relative to femoral length, and shorter tibial lengths relative to femoral length. This suggests that thalattosuchians had—compared to modern crocodylians—proportionally larger skulls, shorter femora, smaller forelimbs, and shorter tibiae, perhaps in relation to a marine lifestyle. This contention is supported by the fully pelagic metriorhynchids having even shorter humeri and tibiae relative to the femora than teleosauroids (Fig. 13).

When we examine the H:F ratio, *Aeolodon priscus* (H:F ~0.36) falls very close to the range of pelagic metriorhynchids (H:F ~0.20–0.34), and outside the range of any other known teleosauroids (H:F ~0.38–0.76) (Fig. 13). In comparison, in extant crocodylians the H:F ratio is considerably higher (H:F ~0.76–0.97), as *Crocodylus, Alligator, Gavialis, Mecistops, Caiman, Melanosuchus, Paleosuchus, Osteolaemus,* and *Tomistoma* have humeri and femoral lengths closer to subequal. The Toarcian teleosauroids have a H:F ratio range of 0.43–0.76, whereas the Callovian species have a range of 0.38–0.52, suggesting a generalised trend towards shortening the humeri in Teleosauroidea across time (with the Late Jurassic *Aeolodon priscus* being the most extreme known example of this trend).

For the T:F ratio, *Aeolodon priscus* (H:F ~0.40–0.46) is intermediate between the ranges of pelagic metriorhynchids (H:F ~0.27–0.43) and other known teleosauroids (H:F ~0.41–0.70) (Fig. 13). In comparison, in extant crocodylians the T:F ratio is again higher (T:F 0.67–0.81). The Toarcian teleosauroids have a T:F ratio range of ~0.48–0.70, the Callovian species have a range of 0.43–0.51, and the three known Kimmeridgian-early Tithonian specimens complete enough to include in our dataset have a range of 0.40–0.46. Once again, there is a generalised trend towards shortening the tibia over time in Teleosauroidea.

Interestingly, these temporal trends independently occurred in the two subclades of Teleosauroidea. The basal Toarcian teleosauroid *Steneosaurus gracilirostris* has high ratios (H:F = 0.685, T:F = 0.687). In the 'T'-subclade, the H:F ratio decreases throughout the Jurassic: Toarcian 0.683–0.761 (*Platysuchus*), Callovian 0.506–0.528 (*Mycterosuchus*), and Kimmeridgian 0.36 (*Aeolodon*). In the 'S'-subclade, the H:F ratio similarly decreases: Toarcian 0.436–0.640 (*S. bollensis*), and Callovian 0.384–0.479 (*S. leedsi*, *S. edwardsi* and *Lemmysuchus*). Similarly, for the T:F ratio, the 'T'-subclade lowers through the Jurassic: Toarcian 0.610–0.697 (*Platysuchus*), Callovian 0.500–0.510 (*Mycterosuchus*), and Kimmeridgian 0.397–0.464 (*Aeolodon*). This also occurs in the 'S'-subclade: 0.481–0.643 (*S. bollensis*), Callovian 0.434–0.511 (*S. leedsi*, *S. edwardsi* and *Lemmysuchus*), and Kimmeridgian 0.412 (*Machimosaurus mosae*).

Overall three independent lines of evidence—reduction of ornamentation, modification of the appendicular skeleton, and their recovery in deeper-water sediments—hint that by the Late Jurassic, some teleosauroids were beginning to evolve a more pelagic lifestyle. All three lines of evidence are present in *A. priscus*, whereas two of them are clear in *B. megarhinus* (reduced ornamentation, recovery in deeper-water sediments; the relevant postcranial bones to examine for limb reduction are not preserved in any known specimen). As these two taxa are united in a clade, we hypothesize that this lineage of teleosauroids

evolved from nearshore, lagoonal forms and entered deeper waters, modifying their skeletons as they did so.

Interestingly, this pattern may be a response to the deepening of waters in the Late Jurassic. *Foffa, Young & Brusatte (2018)* showed that teleosauroids and other shallow water taxa underwent a morphological and species diversity decline in the Jurassic Sub-Boreal Seaway (JSBS) across the Middle-Late Jurassic boundary, in concert with deepening of local and global sea-levels. In fact, the whole JSBS marine reptile assemblage and ecological niches changed in concert with changing habitats. The diverse array of Middle Jurassic shallow water taxa (longirostrine pliosaurids, teleosauroids, metriorhynchines) declined, and was replaced by an assemblage better suited to higher sea-levels (*Pliosaurus*, geosaurine and ophthalmosaurid radiations) (*Young et al., 2012*; *Benson & Druckenmiller, 2014*; *Foffa, Young & Brusatte, 2018*). Accordingly, the unique body plan of *Bathysuchus* and *Aeolodon* lineage, could be the result of habitat change, and the attempt of a divergent group of teleosauroids to adapt to a new type of environment.

Nevertheless, to the extent of our knowledge, teleosauroids never became fully pelagic completing the land-to-sea transition, unlike their relatives the metriorhynchoids. The latter are a textbook example of a secondary adaptation to an aquatic lifestyle, as witnessed by the numerous osteological (e.g., enlarged skull relative to body length, hypocercal tail, modified limb proportions, paddle-like forelimbs, streamlined bodies, complete loss of osteoderms), and soft tissue adaptations (e.g., enlarged nasal exocrine glands, hypertrophied cranial venous systems, simplified and reduced endocranial sinuses) (e.g., *Fraas, 1902*; *Andrews, 1913*; *Young et al., 2011*; *Brusatte et al., 2016*; *Herrera, Leardi & Fernández, 2018*). While the *Bathysuchus* and *Aeolodon* subclade only shared a few of those adaptations, it still represents an instance of parallel evolution. This demonstrates that different thalattosuchian lineages were dynamically changing their anatomy, lifestyles and habitats during the Jurassic. This may have more generally characterized Mesozoic marine reptile faunas—with the repeated evolution of different types of morphologies and lifestyles in concert with habitat changes (e.g., *Benson, 2013*; *Neenan et al., 2017*). Finally, more complete specimens (especially from the late Tithonian and Cretaceous) and digital segmentation of the neuroanatomy of these taxa may hold the key to validating or disproving our hypothesis that they were becoming increasingly adapted to a pelagic lifestyle, and offer insights on the mechanisms of secondary land-to-deep-water transitions in tetrapods.

## CONCLUSIONS

Here, we describe a new specimen of '*Teleosaurus*' *megarhinus* (DORCM G.05067i-v), figure and re–evaluate the holotype (NHMUK PV OR 43806) and an additional specimen (the LPP specimen), demonstrate that it is indeed a valid species and establish a new monotypic genus, *Bathysuchus*, for the taxon. *Bathysuchus* shares numerous rostrum characters with a large unnamed sub-clade of teleosauroids (*Steneosaurus brevior*, *Teleosaurus cadomensis*, *Platysuchus multiscrobiculatus*, *Mycterosuchus nasutus* and an unnamed taxon from Eastern Asia). This suite of characteristics falsifies the hypothesis that *B. megarhinus* is a subjective synonym of *S. leedsi* (which lacks all of these characters). Based on the pronounced

reduction in dermatocranial and osteoderm ornamentation in *Bathysuchus* and the closely related *Aeolodon priscus*, we hypothesise that by the Late Jurassic at least one lineage of teleosauroids evolved a more pelagic lifestyle, perhaps in response of sea-level raising. This helps to explain the paradoxical discovery of *Bathysuchus* in the deep-water Dorset succession of the Kimmeridge Clay Formation. Furthermore, in *Aeolodon priscus*, the post-cranium is well known, and shows skeletal evidence for a pelagic shift. This suggests that metriorhynchids were not the only thalattosuchians to transition from nearshore environments to a fully marine habitat, but that some teleosauroids convergently made the same switch. The future assessment of promising and newly available materials from the Late Jurassic of continental Europe (and particularly the re-description of *Aeolodon priscus* and '*Steneosaurus' jugleri*) is likely to shed light over the interrelationships, evolution and ecology of the peculiar T-clade within Teleosauroidea.

**Anatomical abbreviations**

| | |
|---|---|
| **bo** | basioccipital |
| **Dn** | n$^{th}$ dentary alveolus |
| **d** | dentary |
| **en** | external nares |
| **exo** | exoccipital |
| **fr** | frontal |
| **?j** | ?jugal |
| **?lac** | lacrimal |
| **Mn** | n$^{th}$ maxillary alveolus |
| **ms** | mandibular spatula |
| **mx** | maxilla |
| **na** | nasal |
| **otf** | orbitotemproal foramen |
| **oc** | occipital condyle |
| **or** | orbit |
| **par** | parietal |
| **po** | postorbital |
| **pop** | paraoccipital process |
| **Pn** | n$^{th}$ premaxillary alveolus |
| **pmx p.** | premaxillary projection |
| **pmx** | premaxilla |
| **?prf** | ?prefrotnal |
| **pro** | prootic |
| **pt** | pterygoid |
| **q** | quadrate |
| **so** | supraoccipital |
| **sp** | splenial |
| **sq** | squamosal |
| **stf** | supratemporal fossa |
| **XII** | cranial nerve twelve |

**Institutional abbreviations**

| | |
|---|---|
| **BHN** | Musée-sur-Mer Boulogne, France (closed over a decade ago) |
| **BRSMG** | Bristol Museum and Art Gallery, Bristol, England, UK |
| **CAMSM** | Sedgwick Museum, Cambridge, England, UK |
| **DORCM** | Dorset County Museum, Dorchester, England, UK |
| **IVPP** | Institute of Vertebrate Paleontology and Paleoanthropology, Beijing, China |
| **LPP** | Institut de paléoprimatologie, paléontologie, humaine évolution et paléoenvironnements Université de Poitiers, Poitiers, France |
| **MJML** | Museum of Jurassic Marine Life—the Steve Etches Collection, Kimmeridge, England, UK |
| **MNHN** | Muséum national d'Histoire naturelle, Paris, France |
| **MNHNL** | Muséum national d'Histoire naturelle Luxembourg, Luxembourg City, Luxembourg |
| **NHMUK** | vertebrate palaeontology collection of the Natural History Museum, London, |
| **PV** | UK (OR, old register; R, reptiles) |
| **NHMW** | Naturhistorisches Museum Wien, Vienna, Austria |
| **OUMNH** | Oxford University Museum of Natural History, Oxford, England, UK |
| **SMNS** | Staatliches Museum für Naturkunde, Stuttgart, Baden-Württemberg, Germany |

# ACKNOWLEDGEMENTS

DF, MMJ and MTY should be considered equal first co-authors of this project, because they contributed equally in research time, specimen examination, and writing. We thank the Photography Department of Natural History Museum (NHMUK) for providing quality photos of NHMUK PV OR 43086. Thanks to Paul Tomlinson and Heather Middleton for help during DF's visit at Dorset County Museum (DORCM). Thanks to M Riley (CAMSM), X Xu and L Zhang (IVPP), G Garcia and F Guy (LPP), R Allain (MNHN), B Thuy and R Weis (MNHNL), U Göhlich (NHMW), S Maidment (NHMUK) and R Schoch and E Maxwell (SMNS) for MMJ's access to collections. We would like to thank Stéphane Jouve, Eric Wilberg, Attila Ösi, and the Editor Hans-Dieter Sues for their detailed reviews and comments that greatly improved the quality of this manuscript.

## Funding

Davide Foffa's museum visits were funded by the Small Grant Scheme '2015 Wood Award' (PASW201402), the Systematics Research Fund and the Richard Owen Research Fund by the Palaeontographical Society. Michela M. Johnson is supported by a Natural Sciences and Engineering Research Council of Canada grant (PGS D3-487581-2016) and additional museum visits were funded by the Richard Owen Research Fund by the Palaeontographical Society and Small Grant Scheme 'Sylvester Bradley Award', (PA-SW201601). Mark T. Young and Stephen L. Brusatte are supported by a Leverhulme Trust Research Project

grant (RPG-2017-167). Stephen L. Brusatte is also supported by a Marie Curie Career Integration Grant (630652). Davide Foffa is funded by the Royal Commission for the Exhibition of 1851 - Research Fellowship (2018). The funders had no role in study design, data collection and analysis, decision to publish, or preparation of the manuscript.

## Grant Disclosures

The following grant information was disclosed by the authors:
Small Grant Scheme '2015 Wood Award': PASW201402.
Palaeontographical Society.
Natural Sciences and Engineering Research Council of Canada grant: PGS D3-487581-2016.
Palaeontographical Society and Small Grant Scheme 'Sylvester Bradley Award': PA-SW201601.
Leverhulme Trust Research Project grant: RPG-2017-167.
Marie Curie Career Integration Grant: 630652.
Royal Commission for the Exhibition of 1851.

## Competing Interests

Mark T. Young is an Academic Editor for PeerJ.

## Author Contributions

- Davide Foffa conceived and designed the experiments, performed the experiments, analyzed the data, contributed reagents/materials/analysis tools, prepared figures and/or tables, authored or reviewed drafts of the paper, approved the final draft.
- Michela M. Johnson conceived and designed the experiments, performed the experiments, analyzed the data, contributed reagents/materials/analysis tools, authored or reviewed drafts of the paper, approved the final draft, phylogenetic analyses.
- Mark T. Young conceived and designed the experiments, performed the experiments, analyzed the data, contributed reagents/materials/analysis tools, authored or reviewed drafts of the paper, approved the final draft, supervision.
- Lorna Steel contributed reagents/materials/analysis tools, authored or reviewed drafts of the paper, approved the final draft, access to material and photographs.
- Stephen L. Brusatte authored or reviewed drafts of the paper, approved the final draft, supervision.

## Data Availability

The measurements and phylogenetic dataset, including the description of the characters, taxa and scores used in the analyses, are available in the Supplemental Files.

## New Species Registration

The following information was supplied regarding the registration of a newly described species:
Publication LSID: urn:lsid:zoobank.org:pub:BA30BB3C-9D18-48ED-A79B-AA660450E54B
*Bathysuchus* LSID: urn:lsid:zoobank.org:act:5D902DD6-AE09-466D-8C40-C729F8481636.

## Supplemental Information

Supplemental information for this article can be found online at http://dx.doi.org/10.7717/peerj.6646#supplemental-information.

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
