# Peer review of "Revision of the Late Jurassic deep-water teleosauroid crocodylomorph Teleosaurus megarhinus Hulke, 1871 and evidence of pelagic adaptations in Teleosauroidea"

_PeerJ, doi:10.7717/peerj.6646_

## Round 0.1 · original submission · Major Revisions

The three reviewers offer numerous comments and suggestions, which the authors must address point-by-point when preparing a revised version of the manuscript.

·

Basic reporting

This MS describes an anterior end of a rostrum, a tooth, and 3 osteoderms of a teleosauroid crocodylomorph. Though this specimen is quite fragmentary, it possesses important overlapping elements with the type of ’Teleosaurus’ megarhinus described from the same stratigraphical horizon. English of the MS is clear, article structure is ok.
I think this is an important contribution to vertebrate paleontologists working with crocodiles and/or Mesozoic marine vertebrates, so, after some minor changes, I would be happy to see it published.

Experimental design

Research question is well defined, relevant & meaningful, methods are described with sufficient detail.

Validity of the findings

OK

Additional comments

1) The authors refer the new specimen „DORCM G.05067i-v to ‘Teleosaurus’
megarhinus,” and say „that the species is indeed a valid taxon, and establish a new monotypic
genus, Bathysuchus.” I can accept this taxonomical interpretation but in this case, in the systematic paleontology section a differential diagnosis should be added where they conclude why the two Kimmeridge Bay and possibly the Quercy specimens are not Teleosaurus.

2) An important part of the work is the comparison of the new rostrum fragment with that of the Quercy specimen. So, I strongly suggest to add at least a photo from the French specimen, or, rather a comparative drawing of the narial end of the rostra of at least the three different specimens referred to Bathysuchus.

3) A locality map, as part of Figure 1, would be useful.

4) Citation-reference block is very incomplete! Many of the cited works are not in the Reference list, indicated in the annotated pdf (see attached).


5) Some minor comments and typos have been indicated in the annotated pdf.

·

Basic reporting

no comment

Experimental design

no comment

Validity of the findings

no comment

Additional comments

Overview: The authors describe new material of the poorly known teleosauroid species “Teleosaurus” megarhinus, providing it with a new genus name and establishing diagnostic characters for the taxon. In addition to extensively describing the fragmentary material, they note some characteristics that may be related to adaptation to a more pelagic lifestyle. This is interesting because this taxon is one of the only teleosauroids known from deep water deposits (whereas most teleosauroids are known from estuarine/coastal deposits), bolstering the hypothesis that this species may have been evolving towards a more pelagic lifestyle than other members of the group. In general, this is a well written and well-done study. I particularly appreciate the authors’ extensive comparisons of Bathysuchus material with other teleosauroid species. I do have a large number of comments, but these are primarily related to relatively minor changes to figures and text. There is an issue with the phylogenetic analysis (see below) that needs to be rectified, but it doesn’t affect the overall results of this study. I look forward to the publication of this manuscript and the continuing clarification of teleosauroid taxonomy and phylogeny by this working group.


General comments:
The authors make a lot of comparisons to A. priscus. Perhaps it would be helpful to add a figure showing the reduced ornamentation of skull and osteoderms and possibly the reduced limb elements in this taxon? This would help clarify the discussion section on pelagic adaptations (and with some of the comparisons noted in the description).

The authors note in the “systematic paleontology” section under “Holotype” that this specimen is severely diagenetically damaged and partially reconstructed. However, they rarely discuss how this diagenetic alteration may affect the morphological interpretations they are making (e.g., the 8-shape of the external naris looks to be exaggerated by dorsoventral deformation). It would also be helpful to mention which parts have been reconstructed (though this could be solved by modifying the figure – see figure comments below).

While a lot of the comments below will likely sound as if I am disagreeing with the authors’ interpretation of the morphology of some of these specimens, this is not my intent. In most cases I’m not disagreeing, but rather pointing out that some of the characters they emphasize are from deformed specimens or are otherwise difficult to observe (e.g. in specimens preserved in slabs), or in some instances that there is conflict in the published literature about a particular feature. I’m noting these so the authors can critically assess their own interpretations. If they are confident in their assessments, then by all means they should publish as such. Another note that applies more generally to the description and comparisons with other taxa – I think the authors should be careful to distinguish when their statements are based on direct observation (i.e., character is clearly visible on a specimen) vs. an interpretation (e.g., “we believe this specimen had this character, though it has been altered by deformation”).

On the online supplementary file: The title used in the online supplementary file differs from the title of the manuscript. The order of authors also differs. Additionally, there are some “track changes” comments remaining in the document.

Issues with the phylogenetic data matrix: I think the data file posted is not the one analyzed for this study. The manuscript says the matrix consists of 142 taxa. However, while the one currently posted does have 142 OTUs, it has Bathysuchus as 2 separate OTUs (the second one presumably being the one adding the Quercy specimen information). Additionally, the taxon Steneosaurus meretrix is included in the data file but is not present in Fig. 6. So, it would seem that the data set analyzed for the manuscript actually has 140 taxa (if S. meretrix is not supposed to be included and excluding one Bathysuchus specimen for each analysis) or 141 (if S. meretrix was included but is missing on the figure for some reason). When I exclude the second Bathysuchus OTU, which should give me the first analysis in the manuscript, I get different results than reported. The topology of teleosauroidea is identical to that of Fig. 6, but the tree length is longer (1504 vs 1488 reported) and the number of trees is much larger (14904 vs 102 reported). Excluding Steneosaurus meretrix only reduces tree length by 2. I also think that the authors may not have actually treated their additive characters as additive when they ran the analysis. If I set all characters to unordered, I get a tree length closer to that reported (1486 vs 1488), but the number of trees is still higher (1944 vs 102), and the topology of teleosauroidea is still identical. Running what should be the equivalent of the second analysis (excluding the first Bathysuchus OTU) gives again, a higher tree length (1507 vs. 1491) and a higher number of trees (4968 vs 75). Regardless, the authors should double check their phylogenetic analyses and make any changes necessary to their results section. They should also upload a separate data file for each analysis so there is no ambiguity as to how to reproduce their results.


Specific comments:
Line 30: “is” should be “are” (unless UK English typically treats “whereabouts” as singular); also in line 207

Line 83: “Vignaud (1997)” should be 1995

Lines 102-104: “the supertree analysis of Bronzati...” – This statement isn’t entirely true. Bronzati et al. merely included Mueller-Töwe’s tree in their analysis. I don’t know that they considered the “validity” of the taxa sampled in that analysis. Their recovery of S. megarhinus as sister to Teleosaurus isn’t independent of the Mueller-Töwe analysis. I suspect that no other tree included sampled S. megarhinus, so this is the only relationship for this taxon that could have been recovered.

Line 123: I believe a colon is missing following the word “locality”

Line 126: “continue” should be “continues”; “rock” should be “rocks”

Line 130: “concetional” should be “concretional”

Line 132: “Zone” should be “zones”

Line 134: either there needs to be commas offsetting the clause “spanning the Middle-Late Jurassic”, or the word “that” needs to be moved to follow this clause.

The boilerplate text about ICZN compliance is repeated (lines 145-154 and 187-196)

Lines 213-214: On Platysuchus having 5 premaxillary teeth – On which specimen are the authors basing this number? The SMNS holotype specimen and Urweltmuseum Hauff specimen are both preserved in dorsal view in slabs. I’m not sure it is possible to determine the number of alveoli in these specimens (Mueller-Töwe 2006 suggests 3 premax teeth. I have “4?” written in my notes from when I looked at the holotype – though I don’t recall how I got that number). I don’t know of any Platysuchus specimens preserved with the ventral surface of the premax visible (though they very well could exist somewhere). The only Platysuchus specimen referenced that shows a ventral view of the premax is the Luxembourg specimen. However, I’m not sure I’m convinced that this specimen is attributable to Platysuchus. Johnson et al. attribute it based on it having 5 premaxillary teeth and first and second alveoli “similar in size” rather than the “first slightly smaller than the second” as in S. bollensis. The distinction of “similar in size” vs. “slightly smaller” doesn’t seem like a clear diagnostic feature. Again, as I don’t know of any Platysuchus specimens showing the ventral surface of the premax, I’m not sure how this diagnostic feature (or number of premax teeth) was determined in the first place. The Luxembourg specimen also distinctly lacks the lateral expansion of the premaxilla found in the holotype and UH specimens.

Line 219: “anterior and posterior processes of the premaxilla” – I believe the authors are here referring to short processes along the anterior and posterior margin of the external naris. However, “posterior process of the premaxilla” typically refers to the portion extending posteriorly along the dorsal surface of the rostrum between the maxillae. I think the authors should come up with a different term for these projections along the narial border. It would also make discussion of these features later in the text clearer.

Line 221: please give the specimen number for the “Chinese teleosauroid”

Line 223: I think a semicolon is missing after the word “nares”

Line 225: Is the pronounced lateral expansion of the premaxilla is really autapomorphic for this taxon? Platysuchus and the Chinese teleosauroid also have laterally expanded premaxillae (the authors even note in the discussion that a laterally expanded premaxilla is shared by Bathysuchus, Platysuchus, Mycterosuchus, and the Chinese and Thai teleosauroids).

Line 227: missing a comma following “brevior”

Lines 231-232: Are the ornamentation patterns really autapomorphic? If so, how do they differ from other taxa with lightly ornamented maxillae (e.g. Aeolodon, Chinese and Thai forms)?

Line 262: as mentioned above, I’m not sure this MNHNL specimen is actually Platysuchus

Line 263: The specimen number listed for T. cadomensis is of a specimen lacking a snout, so this should probably not be given as evidence for 5 premaxillary teeth.

Lines 273-274: “Overall the external nares constitute a small length of the entire premaxillae that develop posteriorly for more than 67% of its entire length” - I’m not exactly sure what this statement means. Are the authors talking about the posterior process of the premax in relation to the anteroposterior length of the naris in dorsal view? This doesn't seem like a reliable measurement given deformation... If they're referring to something else, this should be made clearer. Based on how this is worded in the diagnosis, I guess what they mean is that the portion of the premaxilla posterior to the external nares is 67% of the total length of the premax?

Line 276: “projections of the premaxillae in anterior and dorsal direction” – similar to the diagnosis, I think the authors should modify how they are describing these processes on the anterior and posterior margin of the naris. Perhaps describing each process individually would alleviate the confusion.

Lines 277-278: “anterior margin of the external nares… ornamented by fine ridges” – Based on Fig. 3E, it looks like these ridges are along the anterior margin of the premaxilla (above alveoli), rather than along the anterior rim of the naris. This also seems to be a common feature of many teleosauroids (e.g. similar ridges are visible in “S.” heberti and Lemmysuchus).

Line 284: “premaxilla” should be “premaxillae”

Line 286: the word “was” is unnecessary

Line 287: “horizontal (square-shaped) profile…” - here, a line interpretation, or sutures drawn over the images would help. Also, "horizontal" doesn't really describe the shape of this suture well. The maxillae send anterior processes between the palatal processes of the premaxillae. The anterior margin of the suture may be horizontal, but the overall suture must be U-shaped (with a straight anterior margin), or something like that.

Lines 289-290: on the weak ornamentation of the premaxilla – It seems to me that most teleosauroids lack pronounced ornamentation on the premaxilla, with M. nasutus being an outlier.

Line 292: I don’t think the parenthetical “s” in “suture(s)” is really necessary. It is safe to assume the maxillae contact more than one element.

Lines 300-301: on the statement that M. nasutus, A. priscus, and Platysuchus are the closest relatives of Bathysuchus – based on the phylogeny presented in this paper, A. priscus is not closely related to Bathysuchus.

Lines 304-306: “snout length is a plastic feature… independently diverged towards extreme longirostrine or mesorostrine/brevirostrine morphologies (see Discussion)” – I don’t see the topic of snout length evolution anywhere in the Discussion section. Either this should be added (which could be interesting), or this reference to the Discussion should be removed.

Line 317: the word “of” is missing following “most”

Lines 318-319: “spacing is as wide as adjacent alveolar length or more” – In the diagnosis, the authors state this spacing is “longer than adjacent alveoli”. I don’t know which of these statements is accurate, but they should match.

Line 324: “beginning of the posterior end of the snout” – How are the authors defining the posterior end of the snout? The anterior margin of the orbit?

Lines 324-325: “…roughly corresponding to the M27-M30 alveoli” – the authors should note somewhere (possibly near the beginning of this paragraph) that the Quercy specimen is missing the posterior-most alveoli and that M30 does not correspond to the end of the tooth row. Prior to looking at the figure in Vignaud et al., I thought this was saying that the tooth row ended 5 cm anterior to the posterior end of the snout (because it is stated that the specimen has ~30 maxillary alveoli), which obviously would be very odd.

Line 336: “guilds” should be “guild”; “as well as” should be “along with”

Lines 345-346: “medial keel that is not present in the other two osteoderms” – It appears the other dorsal osteoderm is broken, missing the lateral(?) edge. Is it possible the keel would have been on this broken portion? Presence or absence of keels is pretty variable in teleosauroids (even along an individual osteoderm series), so it’s hard to make an argument based on phylogeny. Also, that the inferred ventral osteoderm lacks a keel isn’t surprising. I don’t know of any keeled ventral osteoderms.

Lines 354-362: There is a lot of seemingly unnecessary background info about the matrix presented here. I think this paragraph could be cut down substantially as the important information about the matrix is in the following paragraph.

Line 371: “OUT” should be “OTU"; also, please include the specimen number for the Chinese teleosauroid.

Lines 377-394: please include information about which collapsing rule was enforced in TNT somewhere in this paragraph.

Line 397: “H+Y phylogenetic analysis” – This is a trivial point, but I’m not sure it’s necessary to refer to this as “H+Y” since the authors aren’t using any other datasets in this paper (unlike in other papers where they use multiple datasets and referring to them by name makes more sense)

Line 410: “continued to be in a stable…” – I’m not sure what this means.

Lines 417-418: “Postosuchus kirkpatricki lies outside the clade that unites all other taxa…” – I don’t think this is worth mentioning as this relationship is forced by designating this taxon as the outgroup and is not really a result of the analysis.

Line 423: the word “both” is repeated twice in this sentence.

Line 455: “location” should be “locate”

Line 456: “anteroposterior” should be “anteroposteriorly”

Lines 480-482: sorry to keep harping on this number of premaxillary teeth issue, but the published literature isn’t great on this. If the authors have observed 5 premaxillary alveoli in the holotype of Platysuchus, maybe they could note this as a “personal observation”? As for T. cadomensis, this also seem contentious. I don’t have access to Lamouroux 1820, but Westphal 1961 doesn’t seem to mention the number of premax teeth. Westphal 1962 does say 5, but Vignaud 1995 says 4… Is there a specimen number that could be linked with this observation?

Lines 483-485: on the distribution of premaxillary alveoli in M. nasutus – I am unfamiliar with this specimen (NHMUK PV R 3577). I thought the holotype was the only M. nasutus specimen at NHMUK (though I’m probably wrong given who the authors of this paper are and their familiarity with the NHMUK collections). However, the holotype doesn’t preserve the anterior portion of the premaxilla. Has this specimen been published as Mycterosuchus anywhere?

Lines 486-489: on the presence/absence of the processes of the narial border among teleosaurids – I think these projections have a more complex distribution than is described here. The process on the anterior margin seems common (to a greater or lesser degree) to most 3D preserved teleosauroids (e.g. S. heberti, S. brevior; S. bollensis [the less crushed ones]; M. buffetauti); a small posterior border projection is present in the Chinese teleosauroid (and the anterior margin of the naris is poorly preserved, so it is unclear whether it had an anterior projection or not); the anterior projection is also present in the Thai teleosauroids that I’ve seen (though I'm not sure about the specimens the authors looked at).

Lines 493-496: on the presence of ventrally deflected premaxillae in the Chinese teleosauroid and Platysuchus - seems hard to say for Chinese specimen (probably deformed in this region, but the premaxillary alveoli are in line with the maxillary alveoli); both Platysuchus specimens (SMNS holotype and UH specimen) are dorsoventrally crushed, so the actual ventral deflection is hard to estimate (and in general, this is an issue for most teleosaurs that have been crushed...) – though I think there is real variation here.

Lines 497-498: on the direction of the external nares in teleosauroids – this feature is, in my opinion, highly influenced by dorsoventral compression (i.e., the crushing lowers the posterior margin in line with the anterior margin, making the naris appear dorsally oriented). In all 3D preserved teleosauroids of which I’m aware, the nares face anteriorly/anterodorsally. There may be a subtle distinction between anterior and anterodorsal, but I don’t know of any that I would say are dorsally directed.

Line 517: I think “strongly” is probably too strong of a word for describing the ornamentation of most teleosaur snouts. Slight grooves seem like the most common ornamentation among teleosauroids, though Bathysuchus may be less ornamented than average.

Line 521: “thoracic osteoderms of B. megarhinus are not keeled” – this statement is based on a single damaged thoracic osteoderm. Maybe the authors could show a side-by side comparison of the Bathysuchus osteoderm with an Aeolodon osteoderm to help demonstrate the differences?

Lines 526-529: the linking of reduced dermatocranial ornamentation to a pelagic lifestyle – I don’t have an issue with making this connection and it fits nicely with the occurrence of this taxon in deeper water depositional settings. In the diagnosis the authors note that an “inconspicuously ornamented maxilla dorsal surface” is shared with Aoelodon (another putatively more pelagic teleosaurid. However, this is also shared with the Chinese and Thai teleosauroids, both of which come from freshwater fluvial/lacustrine environments, where a pelagic adaptation doesn’t seem a likely explanation. Again, I think the interpretation of reduced ornamentation on the cranium and osteoderms of Bathysuchus as an adaptation for a more pelagic lifestyle is appropriate and important for the authors to emphasize, but they should also note the discrepancy of the occurrence of reduced ornamentation in what would presumably have been the least pelagic of teleosauroids (based on habitat).

Lines 534-535: “reduce the size and thickness of their osteoderms, as well as lose or heavily reduce their ornamentation” – neither taxon actually lose ornamentation altogether (right? I haven’t seen the Frenche A. priscus specimen). Also, I don't believe the authors discussed reduction in size or thickness of B. megarhinus osteoderms in the description. They should add a line or two about this to the description.

Line 537: on the small osteoderms and short tibiae of A. priscus – have these features been discussed in any publications? If not, maybe the authors could include some values (e.g. tibia vs femur length, or something like that), or comparisons with other taxa.

Line 538: on the forelimb of A. priscus becoming flipper-like - are they proportionally shorter or more reduced than other later teleosaurids like S. leedsi or S. durobrivensis (both of which have reduced forelimbs relative to earlier taxa - See Wilberg 2015, but ignore the erroneous placement of Pelagosaurus as a teleosaurid... d’oh)

Line 541: on Brachysuchus coming from deep-water ecosystems – not really important since the two British specimens are from deep-water deposits, but did the Quercy specimen also come from an inferred deep-water deposit?
Line 550: maybe include the Pierce et al (2009) citation for the hypothesis that B. megarhinus was a subjective synonym of S. leedsi here.

Lines 553-554: “B. megarhinus had evolved a pelagic lifestyle” – maybe “more pelagic” would be a better term here. While Brachysuchus has some features suggestive of a more pelagic lifestyle than other teleosauroids, these features don’t come close to approaching the pelagic adaptations of metriorhynchoids.


Comments on figures:
** It looks like most of my issues with the figures were related to the “auto gamma correction” used for the reviewing manuscript – the original image files look much better.
Fig. 2:
- I feel that parts E and F should be rotated 90 degrees so the dorsal surface is oriented up and ventral down.
- I also feel this figure could be greatly improved by either adding line interpretations (for dorsal, ventral, and one of the lateral views), or at least highlighting which portions of the specimen have been reconstructed.
- I also think it would help for the authors to include a close-up view of the ventral surface of the premaxillae as the premaxillary tooth count and alveolar arrangement are important diagnostic characters referenced often in the text.
Fig. 2 caption: “premaxilla” should be “premaxillae”

Fig. 3:
- This figure would also benefit from either line interpretations or drawing inferred sutures on the photographs themselves.
- Part E should be rotated 90 degrees as recommended for fig. 2
- Part C – is the p5 alveolus visible on this specimen? Again, line interpretations would help.

Fig. 5 caption: E and F are identified as “?ventral” osteoderms, but in the text this is definitively identified as ventral.

Fig. 6:
- What is the dashed branch leading to “Sphenosuchians”? I’m assuming this is a stand in for a paraphyletic grade. This should be noted in the figure caption.
- The clade labeled “Crocodyliformes” should be “Metasuchia” (Crocodyliformes would be the node bounding this clade + Thalattosuchia & Eopneumatosuchus)
- There is an aberrant apostrophe preceding “Chinese teleosauroid” in both parts A and B

·

Basic reporting

The paper provides a revision of the teleosauroid species T. megarhinus, providing new material. The original material is poor (an anterior portion of a snout), and the new material is also poor (an anterior extremity of a snout). The authors consider that T. megarhinus is a valide species and erect a new genus name.
The description is good and well itemized.

Experimental design

'no comment'

Validity of the findings

If the description is good and well itemized, I strongly disagree with the characters used in the diagnosis as with some from the comparison. I agree that T. megarhinus is probably a valid species, supported by some characters, but these characters are fewer that suggested in the paper. Most of those used to diagnose the new genus are not clear and disputable (I discus that in detail below). They can be found in many thalattosuchians. Most of them are related to the shape of the anterior part of the snout, but this part is often strongly deformed in most of the thalatosuchians specimens and this strongly influence our interpretation of the character. This deformation has not been considered in the paper while it is often strong in thalattosuchians (in particular in specimens from Oxford Clay (Vignaud, 1995)). The specimens presented here are also deformed. If some characters suggest the validity of the species, I think that erect a new genus name is too daring. First, it is mainly based on two fragments of anterior extremity of two snouts (it was not possible to provide comparison with a lost complete French skull), and second, the teleosauroid phylogeny is particularly unclear, and the relationships of T. megarhinus could strongly varies in the future, the poor preservation of the remains (only anterior portion of snout) will probably increase this instability.
The result of the phylogenetic analysis and relationships of B. megarhinus should be presented with more carful du to the poorness of the specimens.
I invite the authors to develop and precise the morphological points I discussed below to eliminate all possible doubt (as mine).
I detailed my comments on the morphology in the pdf file.

The authors also discussed a possible pelagic adaptation in Teleosauroidea based on the reduced ornamentation in B. megarhinus. The ornamentation is weak in almost all longirostrine Crocodylomorphs, in particular in the snout area. Poor or no ornamentation is seen in S. pictaviensis (Vignaud, 1998), S. durobrivensis, S. heberti, S. larteti.
Ornamentation of osteoderms with widely separated pits is also observed in T. cadomensis. The observation in B. megarhinus is based on a unique incomplete osteoderm. Some degrees of variation exists in the ornamentation of the osteoderms within the same species, and some time within the same individual, and the distance between the pits increases with the age (see Mueller-Töwe, 2006; Vignaud, 1998)(CF. German S. bollensis).
I am not sure that the difference in the ornamentation of the snout (maxilla) and osteoderms between teleosauroids indicate more pelagic adaptation in some species of the group, I think the differences are too weak.

---

## Round 0.2 · Minor Revisions

Please respond to the reviewer's comments and modify the manuscript where necessary.

·

Basic reporting

no comment

Experimental design

no comment

Validity of the findings

no comment

Additional comments

Comments on the manuscript “A revision of the deep-water teleosauroid crocodylomorph Teleosaurus megarhinus Hulke, 1871 and evidence of pelagic adaptations in Teleosauroidea”.


I review the first version of the present paper and the authors consistently increased the quality of the paper. They answer to many of the comments I provided in my first review, and bring a particularly interesting analysis on the postcranial morphometry of teleosauroids and their possible adaptation to pelagic environment.
I still have some comments and disagreements, not on the characters, but on the interpretation of some of them in some crushed specimens.

Detailled comments

Diagnosis: “the premaxillae have five alveoli (shared with Platysuchus multiscrobiculatus Berckhemer, 1929, Teleosaurus cadomensis Lamouroux, 1820)”: 5 shared with a number of other teleosauroids.: shared with other thalattosuchians such as in S. deslongchampsianus from Kimmeridgian of France, and S. priscus from Germany (See Godefroit et al., 1995). They also must be cited, considering the provided diagnosis, only Platysuchus multiscrobiculatus and Teleosaurus cadomensis share the presence of 5 pmx teeth.

“in dorsal view the external nares have an ‘8’ shape, created by the enlarged anterior and posterior projections of the of the premaxilla (shared with Mycterosuchus nasutus);”: We agree that the anterior projection of the premaxilla is present in many teleosauroids. Yet this is not the unique feature that contribute to give the external nares a ‘8’-shape. Furthermore our new comparative figures show an array of teleosauroids and demonstrate that only a restrict sub-set has ‘8’-shape external nares. We agree that this is not a unique feature of Bathysuchus. The impact of preservation on this character has also been carefully considered. Even in dorsoventrally flattened skulls all these characters can still be confidently assessed in the majority of cases where the whole snout is preserved.”: OK, but it is also probably shared with T. cadomensis (see Eudes-Deslongchamps, 1870; Brignon, 2014) and S. megistorhynchus (see Godefroit et al., 1995).

“created by the enlarged anterior and posterior projections of the of the premaxilla”

“reduced antroposterior length of the external nares”: anteroposterior

“more than 67% of the premaxillae total length is posterior to the external nares [shared with ‘Steneosaurus’ gracilirostris Westphal, 1961 and the Chinese teleosauroid (IVPP V 10098)];” : Also in T. cadomensis (68%), S. megistorhynchus (80%), S. brevior (80%), P. multiscrobiculatus (69%), S. obtusidens (68%, from fig from Andrews), and probably S. larteti.

P14: “in fact, the P1 and P2 alveoli are laterally aligned, but are also well-separated.”: If the authors consider that the alveoli are not nearly confluent as observed in S. leedsi and S. edwardsi, it cannot be said that they are well separated. Moreover, the difference between both character states is weak, and I am not sure that the separation between P1 and P2 differs significantly between cited species.

P15: “Yet, it is noticeably more extreme in Bathysuchus megarhinus than in the other taxa. It is possible that this condition has been slightly exaggerated by the preservation of the specimens. However, we see no reason why a dorsoventral compaction would result in moving the premaxillae downwards rather than buckling it on the same level as the maxillae”: Lateral expansion and ventral deflection of the pmx: we agree and acknowledge that the pmx is laterally expanded in other teleosaurid. However we added a comparative table to show that the extent of the lateral expansion of the pmx is particularly exaggerated in Bathysuchus specimens. Similarly for the ventral deflection. We added further discussion and comparisons for both these characters”: The problem is not that the ventral deflection is exaggerated by dorsoventral flattening in B. megarhinus (I agree with the authors), it is that the ventral deflection could be decreased by this flattening in other species for which the specimens are strongly flattened. It is the case in many species. I agree with the authors that the ventral deflection is probably stronger in B. megarhinus, but this character is difficult to evaluate (and thus to compare) in many teleosauroid species.

---

## Round 0.3 · accepted · Accept

With all reviewers' comments fully addressed, the revised version of the manuscript will be recommended for acceptance for publication.

#